# MILR: Improving Multimodal Image Generation via Test-Time Latent Reasoning

**Yapeng Mi**[1,2]    **Yanpeng Zhao**[2] † ✉    **Hengli Li**[2,3]    **Chenxi Li**[2,4]

**Huimin Wu**[2]    **Xiaojian Ma**[2]    **Song-Chun Zhu**[2,3]    **Ying Nian Wu**[5]    **Qing Li**[2] ✉

[1]University of Science and Technology of China
[2]State Key Laboratory of General Artificial Intelligence, BIGAI
[3]Peking University    [4]Tsinghua University    [5]University of California, Los Angeles
⊕ **Project:** https://spatigen.github.io/milr.io/    ⬠ **Code:** https://github.com/spatigen/milr

## ABSTRACT

Reasoning-augmented machine learning systems have shown improved performance in various domains, including image generation. However, existing reasoning-based methods for image generation either restrict reasoning to a single modality (image or text) or rely on high-quality reasoning data for fine-tuning. To tackle these limitations, we propose MILR, a test-time method that jointly reasons over image and text in a unified latent vector space. Reasoning in MILR is performed by searching through vector representations of discrete image and text tokens. Practically, this is implemented via the policy gradient method, guided by an image quality critic. We instantiate MILR within the unified multimodal understanding and generation (MUG) framework that natively supports language reasoning before image synthesis and thus facilitates cross-modal reasoning. The intermediate model outputs, which are to be optimized, serve as the unified latent space, enabling MILR to operate entirely at test time. We evaluate MILR on GenEval, T2I-CompBench, and WISE; it achieves state-of-the-art results on all benchmarks. Notably, on knowledge-intensive WISE, MILR attains an overall score of 0.63, improving over the baseline by 80%. Our further analysis indicates that joint reasoning in the unified latent space is the key to its strong performance. Moreover, our qualitative studies reveal MILR's nontrivial ability in temporal and cultural reasoning, highlighting the efficacy of our reasoning method.

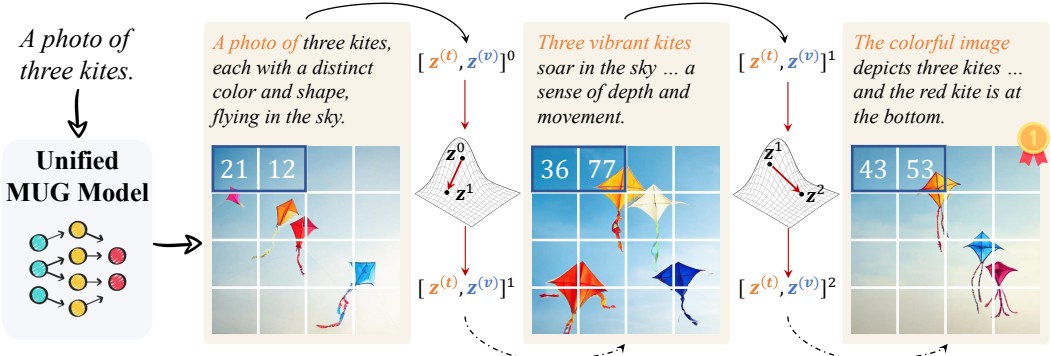

Figure 1: Latent reasoning of MILR. The black *solid* line denotes extracting the output vector representations $\mathbf{z}^k$ of the text tokens $\mathbf{z}^{(t)}$ and image tokens $\mathbf{z}^{(v)}$ to be optimized, and the black *dashed* line denotes decoding from the optimized latent vectors $\mathbf{z}^{k+1}$, where $\mathbf{z} = [\mathbf{z}^{(t)}, \mathbf{z}^{(v)}]$.

---

Contact: `miyapeng78@gmail.com`, `yannzhao.ed@gmail.com`, `dylan.liqing@gmail.com`.
†: Project Lead.  ✉: Corresponding Author.

# 1 INTRODUCTION

Text-guided image generation is the task of synthesizing an image conditioned on a given text instruction. In recent years, the field has witnessed transformative progress: moving from generative adversarial models (Goodfellow et al., 2014) to autoregressive and diffusion approaches (Sun et al., 2024; Esser et al., 2024; Black Forest Labs, 2024; Chen et al., 2025; Wu et al., 2025a). However, traditional models are limited in generating images in a single-shot fashion and thus are unable to resolve potential defects (Huang et al., 2025a; Niu et al., 2025). Inspired by the success of reasoning-augmented LLMs—such as OpenAI o1 (OpenAI et al., 2024) and DeepSeek-R1 (DeepSeek-AI et al., 2025)—that can reflect on and refine their thoughts, and answer accordingly, recent works have attempted to endow image generation models with reasoning ability (Guo et al., 2025; Fang et al., 2025; Zhang et al., 2025b; Wu et al., 2025b; Chern et al., 2025).

Reasoning-augmented image generation models typically perform reasoning in two spaces: language and image spaces. Language reasoning primarily involves refining instructions to make them more comprehensible to the model (Wu et al., 2025b; Li et al., 2025b), and image reasoning iterates on the generation guided by a quality metric (Guo et al., 2025; Zhuo et al., 2025). Early works implement reasoning either in the language or in the image space, lacking a mechanism for synergistic reasoning across the two spaces. To fill the gap, recent works resort to unified multi-modal understanding and generation (MUG; Jiang et al. (2025a); Duan et al. (2025); Zhang et al. (2025b)), which natively supports language reasoning before generating images (referred to as multimodal image generation), facilitating cross-modal image-text reasoning. Despite the success, these approaches require carefully curated reasoning data and depend on model fine-tuning, rendering them complex and costly to develop in practice.

To address these limitations, we propose Multimodal Image generation via test-time Latent Reasoning, dubbed MILR. Our core idea is to reason in a unified *latent* vector space that encodes both text and images, starkly different from previous methods that explicitly reason over raw images and text. We build MILR upon a Transformer-based MUG model and choose the unified latent space represented by the intermediate output vectors. Since the shared latent space is modality-agnostic, it provides a unified view of multimodal reasoning, reducing the modality gap and improving the overall efficacy of cross-modal reasoning.

Reasoning with MILR involves finding the best vector representations of image and text tokens that lead to improved image quality. We implement it using the policy gradient method (Williams, 1992), where, at test time, the reward is computed by scoring the compatibility between the generated image and the given instruction. Crucially, gradients are only back-propagated to the cross-modal latent representations (i.e., the intermediate model outputs) obtained from the forward pass (see Figure 1), without altering any model parameters, and thus making MILR a test-time latent reasoning method.

We evaluate MILR on three widely-used image generation benchmarks: GenEval (Ghosh et al., 2023), T2I-CompBench (Huang et al., 2023), and WISE (Niu et al., 2025). MILR established new state-of-the-art results across all benchmarks. Notably, it achieves an overall score of 0.95 on GenEval, matching the best training-based model and outperforming the best test-time-scaling method by 4.4%. On the more challenging WISE benchmark, MILR obtains an overall score of 0.63, surpassing the strongest baseline model by 16.7%. Our further analysis reveals that the best performance of MILR is driven by its ability to perform joint image-text reasoning in a unified latent space. This unique advantage also makes MILR successful in instructions that involve challenging cultural and temporal reasoning.

To summarize:

- We introduce MILR, a test-time reasoning-augmented method that improves image generation by performing joint image-text reasoning in a unified latent space.

- We demonstrate the effectiveness of MILR by showing that it achieves superior performance across three different benchmarks for image generation.

- We conduct a comprehensive analysis, perform various ablation studies, and discuss potential limitations of MILR.

## 2    RELATED WORKS

**Reasoning-Augmented Image Generation.** Remarkable progress has been achieved in reasoning-augmented LLMs (Wei et al., 2022; DeepSeek-AI et al., 2025), but effectively integrating reasoning into conventional text-guided image generation models remains a challenge (Betker et al., 2023; Sun et al., 2024; Esser et al., 2024; Zhao et al., 2024; Chen et al., 2025). With the development of unified multimodal understanding and generation that can generate images and text in an interleaved manner, many works have explored reasoning for text-guided image generation. The predominant paradigm uses causal language modeling to unify image and text generation as next token prediction, enabling the model to make plans via chain-of-thought before generating images (Fang et al., 2025; Deng et al., 2025; Xiao et al., 2025; Cai et al., 2025; Guo et al., 2025; Jiang et al., 2025a; Duan et al., 2025). Another line of research focuses on reasoning over images via the test-time scaling strategy (Li et al., 2025b; Zhuo et al., 2025). These works typically rely on an external critic model to provide feedback on further improvement. Unlike all these methods that explicitly perform reasoning over raw images and text, we use test-time optimization to refine the latent representations of image and text tokens, leading to a unified cross-modal latent reasoning method.

**Reasoning in The Latent Space.** Differently from explicit reasoning via a chain of thoughts (Wei et al., 2022), latent reasoning refers to an implicit reasoning mechanism applied to the latent states (e.g., intermediate outputs) of the model. As in Transformer-based models, latent reasoning is typically implemented as spatial and temporal recurrences. Spatial recurrences implicitly deepen the model by iterating over the latent states between the Transformer layers (Hao et al., 2024; Cheng & Van Durme, 2024; Zhang et al., 2025a; Shen et al., 2025), while temporal recurrences refine the latent states through iterations across input tokens (Dao & Gu, 2024; Geiping et al., 2025). Under the recurrences is the idea of scaling up test-time computation for inference; however, this requires the recurrent modules to be pre-trained. In a similar vein to those, we iterate over the unified image-text latent states only at test time, without introducing and updating any model parameters.

**Reinforcement Learning for Reasoning.** Reinforcement learning (RL) has been the key to eliciting the reasoning ability of large language models (DeepSeek-AI et al., 2025; OpenAI et al., 2024). Inspired by the success of GRPO, which is an improved RL algorithm over PPO (Schulman et al., 2017) and is used in DeepSeek-R1 (Shao et al., 2024), researchers have extended it to the multimodal understanding domain, including visual question answering (Huang et al., 2025b; Liu et al., 2025b). In image generation, RL has also been shown to be effective (Jiang et al., 2025a; Tong et al., 2025; Jiang et al., 2025b; Pan et al., 2025b; Duan et al., 2025; Pan et al., 2025a; Liu et al., 2025a; Xue et al., 2025; Xiao et al., 2025), but has been used as a training-time optimization method. Unlike them, we employ a simple algorithm, REINFORCE (Williams, 1992), for test-time optimization.

## 3    METHOD

### 3.1    REASONING-AUGMENTED MULTIMODAL IMAGE GENERATION

We are interested in a framework for reasoning-augmented image generation that natively supports language reasoning before image synthesis. A typical implementation is through unified multimodal understanding and generation (MUG; Chen et al. (2025); Deng et al. (2025)). Suppose the language token sequence $\mathbf{t} := t_{1:M} := t_1, t_2, \ldots, t_M$, the image token sequence $\mathbf{v} := v_{1:N} := v_1, v_2, \ldots, v_N$, and the given instruction $c$, MUG defines multi-modal image generation as an autoregressive generation process:

$$p(\mathbf{t}, \mathbf{v}|c) = \prod_{n=1}^{N} p(v_n|v_{1:n}, \mathbf{t}, c) \prod_{m=1}^{M} p(t_m|t_{1:m}, c), \tag{1}$$

The generation of $\mathbf{v}$ depends on the reasoning via language tokens $\mathbf{t}$, which are generated from $c$. By analogy to textual reasoning, we refer to the generation of $\mathbf{v}$ as visual reasoning. To produce the final image, $\mathbf{v}$ is further passed through a pixel decoder: $V_f \sim p(\cdot|\mathbf{t}, \mathbf{v}, c)$. Since we focus on discrete image tokens produced by a pre-trained discrete VAE (Chen et al., 2025), the pixel decoder $p(\cdot|\mathbf{t}, \mathbf{v}, c)$ becomes deterministic given $\mathbf{v}$.

Let $R(V_f, c)$ denote a reward model that scores the compatibility between the given instruction $c$ and the final image $V_f$, the goal of test-time reasoning-augmented image generation is to find an

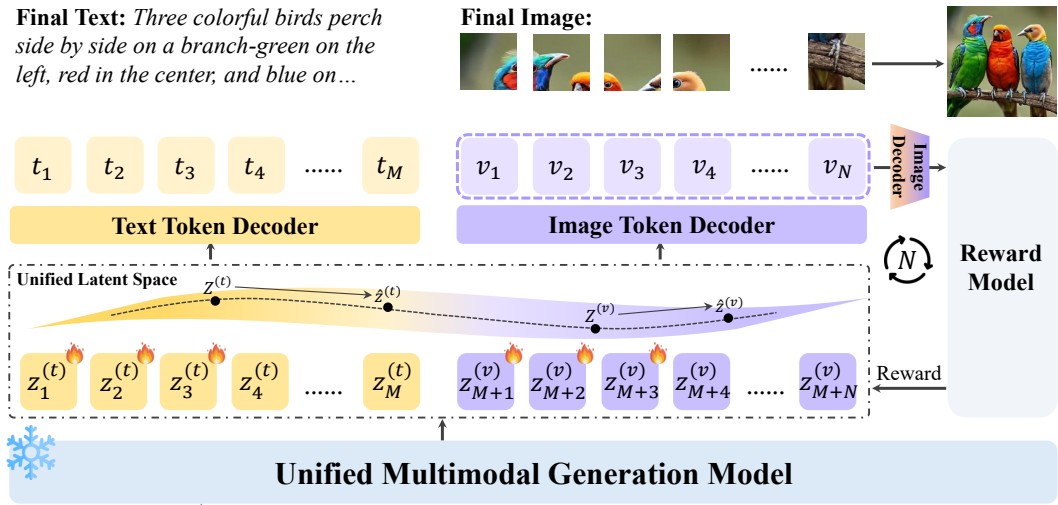

Figure 2: Overview of MILR. MILR performs test-time latent reasoning in a unified latent space; it uses policy gradients to iteratively refine text & image latents $\mathbf{z}^{(t)}, \mathbf{z}^{(v)}$, guided by a reward model. The reward model scores each generated image conditioning on the instruction.

optimal pair $(\mathbf{t}^*, \mathbf{v}^*)$ that maximizes the expected reward under $p(\cdot|\mathbf{t}, \mathbf{v}, c)$, without modifying any model parameters:

$$\mathbf{t}^*, \mathbf{v}^* = \arg\max_{\mathbf{t}, \mathbf{v}} \mathbb{E}_{V_f \sim p(\cdot|\mathbf{t}, \mathbf{v}, c)}[R(V_f, c)]. \tag{2}$$

In the case of MUG, the model itself can act as the reward function because of its multimodal understanding ability; alternatively, any off-the-shelf model that has such a capability can serve the same role (see discussion in Section 4.4). Due to the infinite search space, the problem defined by Equation 2 is generally intractable.

## 3.2 IMAGE GENERATION VIA TEST-TIME LATENT REASONING

### 3.2.1 MULTI-MODAL LATENT REASONING

Rather than searching over discrete image and text tokens, we propose searching in the unified latent vector space, that is, searching over their continuous vector representations. As in MUG, these vectors correspond to the intermediate model outputs at respective token positions. They lie in a vector space that encodes both image and text tokens, offering a unified view of visual and textual reasoning[1] and facilitating cross-modal reasoning (see Figure 2).

Formally, denoting the latent representations of image and text tokens by $\mathbf{z}^{(v)} = z_{1:N}^{(v)}$ and $\mathbf{z}^{(t)} = z_{1:M}^{(t)}$, respectively, where $z^{(v)}, z^{(t)} \in \mathbb{R}^d$ are the outputs from the same Transformer layer and thus lie in a shared $d$-dimensional vector space, we can rewrite Equation 2 as:

$$\mathbf{z}^* = \arg\max_{\mathbf{z}} \mathbb{E}_{V_f \sim p(\cdot|\mathbf{z}, c)}[R(V_f, c)], \tag{3}$$

where $\mathbf{z} = [\mathbf{z}^{(t)}; \mathbf{z}^{(v)}]$ indicates the multimodal latent representation of the token sequence $[\mathbf{t}, \mathbf{v}]$. We refer to this optimization problem as multimodal latent reasoning.

Given the optimal $\mathbf{z}^\star$ from a specific model layer, to produce the final $V_f$, we need to continue the forward pass until it is decoded into discrete tokens $[\mathbf{t}, \mathbf{v}]$. Thus, the pixel image generation becomes:

$$p(V_f|\mathbf{z}^*, c) = p(V_f|\mathbf{t}, \mathbf{v}, c)p(\mathbf{t}, \mathbf{v}|\mathbf{z}^*), \tag{4}$$

where $p(\mathbf{t}, \mathbf{v}|\mathbf{z}^*)$ represents the remaining forward pass of MUG starting with $\mathbf{z}^*$.

---

[1]We refer to "reasoning" as iterative updates in the latent space rather than text-based chain-of-thought.

### 3.2.2 Gradient-based optimization for latent reasoning

In general, the problem defined by Equation 3 does not admit a closed-form solution, so we resort to REINFORCE (Williams, 1992), a policy gradient optimization method. We note that it has been applied to fully textual reasoning in language tasks (Li et al., 2025a), but we, for the first time, extend it to unified multimodal latent reasoning for image generation.

With REINFORCE, we can formulate the cross-modal optimization process as:

$$\mathbf{z}^{k+1} \leftarrow \mathbf{z}^k + \eta \cdot \mathcal{J}(\mathbf{z}^k), \tag{5}$$

$$\mathcal{J}(\mathbf{z}^k) = \mathbb{E}_{V_f \sim p(\cdot|\mathbf{z}^k, c)}[R(V_f, c)\nabla_{\mathbf{z}} \log(p(\mathbf{t}, \mathbf{v}|\mathbf{z}^k))], \tag{6}$$

where $\eta$ is the learning rate. For efficiency, we choose as $\mathbf{z}$ the outputs of the last Transformer layer, i.e., the inputs to the final modal-specific decoding heads. Moreover, we approximate gradients $\mathcal{J}(\mathbf{z})$ using a single sampled pair $(\mathbf{t}, \mathbf{v})$. The gradients are back-propagated only to the model outputs $\mathbf{z}$, without altering any model parameters, and thus making MILR a test-time reasoning method.

Naively, we would optimize[2] all $M + N$ latents in $z_{1:M+N}$, but searching using only the guidance of a reward model is potentially biased, and it does not leverage the generative capacity of MUG for better exploration; instead, we optimize only the first $\lambda_t M$ (where $\lambda_t \in (0, 1]$) latents for text. After decoding them into discrete tokens, we complete textual reasoning via the standard autoregressive generation conditioned on them. As we will later see, this simple strategy strikes a good balance between efficiency and performance (see Section 4.3.1). For visual reasoning, we adopt a similar strategy and optimize the first $\lambda_v N$ (where $\lambda_v \in (0, 1]$) latents.[3] This is further supported by the observations of Hu et al. (2025): the first few tokens govern the global structure of the image, while the remaining tokens primarily influence high-frequency details.[4]

## 4 Experiments

### 4.1 Experimental Settings

**Benchmarks and Baselines.** We conduct experiments on three benchmarks widely used in image generation: GenEval (Ghosh et al., 2023), T2I-CompBench (Huang et al., 2023), and WISE (Niu et al., 2025). We compare MILR against the following three sets of baselines:

- **Non-reasoning models** refer to the classical image generation models that synthesize images from a given instruction in a single shot, without refining the instruction. We consider diffusion models such as FLUX.1-dev (Black Forest Labs, 2024), DALL·E 3 (Betker et al., 2023), and SD3-Medium (Esser et al., 2024), autoregressive models such as LlamaGen (Sun et al., 2024) and Emu3 (Wang et al., 2024), and hybrid autogressive-diffusion models such as BAGEL (Deng et al., 2025) and GPT-4o (OpenAI, 2025).

- **Training-based reasoning models** refer to the image generation models that acquire the reasoning ability through training, including GoT-R1 (Duan et al., 2025), T2I-R1 (Jiang et al., 2025a), Flow-GRPO (Liu et al., 2025a), and GRPO- and DPO-tuned Janus-Pro (Tong et al., 2025).

- **Test-time reasoning models** refer to the models that admit reasoning through tailored inference strategies, including Reflect-DiT (Li et al., 2025b) and ReflectionFlow (Zhuo et al., 2025), which uses language feedback, and Best-of-N and PARM (Guo et al., 2025), which rely on search.

**Hyperparameters and configurations.** We choose Janus-Pro (Chen et al., 2025), an autoregressive model, as the MUG model. For the portions of text and image tokens that are optimized, we perform grid search on a validation split sampled from GenEval and empirically set $\lambda_t = 0.2$ for text and $\lambda_v = 0.02$ for images (see Figure 5). We use the Adam optimizer (Kingma, 2014), where the learning rate is empirically set to 0.03. For each benchmark, we use its own evaluation toolkit as the reward model, following previous work (Liu et al., 2025a; Jiang et al., 2025a; Tong et al., 2025). We

---

[2]We experimented with different optimization strategies, see details in Appendix A.1.

[3]In each iteration, the text token number $M$ may vary while the image token number $N$ remains unchanged.

[4]Rather than optimizing a prefix of image token sequence, we can optimize a random subset of image tokens, but this strategy proves worse than prefix optimization (see Table 5 in Appendix A.1).

Table 1: Results on GenEval. The best score is in bold and the second best is underlined.

| Method | Single Obj. ↑ | Two Obj. ↑ | Counting ↑ | Colors ↑ | Position ↑ | Attr. Binding ↑ | Overall ↑ |
|---|---|---|---|---|---|---|---|
| *Non-reasoning Models* | | | | | | | |
| LlamaGen (Sun et al., 2024) | 0.71 | 0.34 | 0.21 | 0.58 | 0.07 | 0.04 | 0.32 |
| Emu3 (Wang et al., 2024) | 0.98 | 0.71 | 0.34 | 0.81 | 0.17 | 0.21 | 0.54 |
| FLUX.1-dev (Black Forest Labs, 2024) | 0.98 | 0.79 | 0.73 | 0.77 | 0.22 | 0.45 | 0.66 |
| DALL-E 3 (Betker et al., 2023) | 0.96 | 0.87 | 0.47 | 0.83 | 0.43 | 0.45 | 0.67 |
| SD3-Medium (Esser et al., 2024) | 0.99 | 0.94 | 0.72 | 0.89 | 0.33 | 0.60 | 0.74 |
| BAGEL (Deng et al., 2025) | 0.99 | 0.94 | 0.81 | 0.88 | 0.64 | 0.63 | 0.82 |
| GPT-4o (OpenAI, 2025) | 0.99 | 0.92 | 0.85 | 0.91 | 0.75 | 0.66 | 0.85 |
| *Training-based Reasoning Models* | | | | | | | |
| GoT-R1 (Duan et al., 2025) | 0.99 | 0.94 | 0.50 | 0.90 | 0.46 | 0.68 | 0.75 |
| T2I-R1 (Jiang et al., 2025a) | 0.99 | 0.91 | 0.53 | 0.91 | 0.76 | 0.65 | 0.79 |
| Flow-GRPO (Liu et al., 2025a) | **1.00** | **0.99** | **0.95** | 0.92 | **0.99** | 0.86 | **0.95** |
| ReasonGen-R1 (Zhang et al., 2025b) | 0.99 | 0.94 | 0.62 | 0.90 | 0.84 | 0.84 | 0.86 |
| Janus-Pro-7B(+GRPO) (Tong et al., 2025) | 0.99 | 0.87 | 0.61 | 0.87 | 0.82 | 0.68 | 0.81 |
| Janus-Pro-7B(+DPO) (Guo et al., 2025) | 0.99 | 0.89 | 0.65 | 0.92 | 0.82 | 0.72 | 0.83 |
| *Test-time Reasoning Models* | | | | | | | |
| Reflect-DiT (Li et al., 2025b) | 0.98 | 0.96 | 0.80 | 0.88 | 0.66 | 0.60 | 0.81 |
| ReflectionFlow (Zhuo et al., 2025) | **1.00** | 0.98 | 0.90 | 0.96 | 0.93 | 0.72 | 0.91 |
| Janus-Pro-7B(+Text Enhanced Reasoning) | 0.98 | 0.91 | 0.55 | 0.89 | 0.74 | 0.67 | 0.79 |
| Janus-Pro-7B(+Best-of-N) | 0.99 | 0.96 | 0.89 | 0.93 | 0.92 | 0.80 | 0.91 |
| Janus-Pro-7B(+PARM) (Guo et al., 2025) | **1.00** | 0.95 | 0.80 | 0.93 | 0.91 | 0.85 | 0.91 |
| Janus-Pro-1B (Chen et al., 2025) | 0.98 | 0.82 | 0.51 | 0.89 | 0.65 | 0.56 | 0.73 |
| **Janus-Pro-1B+MILR** | **1.00** | 0.91 | 0.78 | 0.92 | 0.86 | 0.86 | 0.89 |
| Janus-Pro 7B (Chen et al., 2025) | 0.98 | 0.85 | 0.56 | 0.89 | 0.77 | 0.64 | 0.78 |
| **Janus-Pro-7B+MILR** | **1.00** | 0.96 | 0.90 | **0.98** | 0.98 | **0.91** | **0.95** |

Table 2: Results on T2I-CompBench and WISE. The best is in bold, and the second is in underlined.

| Method | T2I-CompBench | | | | | | | WISE |
|---|---|---|---|---|---|---|---|---|
| | Color ↑ | Shape ↑ | Texture ↑ | Spatial ↑ | Non-Spatial ↑ | Complex. ↑ | Overall ↑ | Avg ↑ |
| *Non-reasoning Models* | | | | | | | | |
| PixArt-α Chen et al. (2023) | 0.6690 | 0.4927 | 0.6477 | 0.2064 | 0.3197 | 0.3433 | 0.4465 | 0.47 |
| FLUX.1-dev Black Forest Labs (2024) | 0.7407 | 0.5718 | 0.6922 | 0.2863 | 0.3127 | 0.3703 | 0.4957 | 0.50 |
| DALL-E 3 Betker et al. (2023) | 0.7785 | **0.6209** | 0.7036 | 0.2865 | 0.3003 | 0.3773 | 0.5112 | - |
| SD3-Medium Esser et al. (2024) | 0.8132 | 0.5885 | 0.7334 | 0.3200 | 0.3140 | 0.3771 | 0.5244 | 0.42 |
| Show-o Xie et al. (2024) | 0.5600 | 0.4100 | 0.4600 | 0.2000 | 0.3000 | 0.2900 | 0.3700 | 0.30 |
| BAGEL Deng et al. (2025) | 0.8027 | 0.5685 | 0.7021 | 0.3488 | 0.3101 | 0.3824 | 0.5191 | 0.52 |
| *Training-based Reasoning Models* | | | | | | | | |
| T2I-R1 Jiang et al. (2025a) | 0.8130 | 0.5852 | 0.7243 | 0.3378 | 0.3090 | 0.3993 | 0.5281 | 0.54 |
| GoT-R1 Duan et al. (2025) | 0.8139 | 0.5549 | **0.7339** | 0.3306 | 0.3169 | **0.3944** | 0.5241 | - |
| Janus-Pro-7B(+GRPO) Tong et al. (2025) | 0.7721 | 0.5366 | 0.7317 | 0.2869 | 0.3087 | 0.3697 | 0.5010 | - |
| *Test-time Reasoning Models* | | | | | | | | |
| Show-o + PARM Guo et al. (2025) | 0.7500 | 0.5600 | 0.6600 | 0.2900 | 0.3100 | 0.3700 | 0.4900 | - |
| Janus-Pro-7B(+Text Enhanced Reasoning) | 0.7087 | 0.4419 | 0.5821 | 0.2597 | 0.3072 | 0.3761 | 0.4459 | 0.46 |
| Janus-Pro-7B(+Best-of-N) | 0.7089 | 0.4925 | 0.7089 | 0.3542 | **0.3262** | 0.3721 | 0.4938 | 0.52 |
| Janus-Pro-1B Chen et al. (2025) | 0.3411 | 0.2261 | 0.2696 | 0.0968 | 0.2808 | 0.2721 | 0.2478 | 0.26 |
| **Janus-Pro-1B+MILR** | 0.6066 | 0.2796 | 0.4177 | 0.2796 | 0.2622 | 0.2613 | 0.3512 | 0.40 |
| Janus-Pro-7B Chen et al. (2025) | 0.6359 | 0.3528 | 0.4936 | 0.2061 | 0.3085 | 0.3559 | 0.3921 | 0.35 |
| **Janus-Pro-7B+MILR** | **0.8508** | 0.5117 | 0.6949 | **0.4613** | 0.3078 | 0.3684 | **0.5325** | **0.63** |

conduct all experiments on a single NVIDIA A100 80GB GPU. A discussion on model efficiency can be found in Appendix A.3.3.

## 4.2 MAIN RESULTS

MILR achieves state-of-the-art results on GenEval, one of the most widely-used benchmarks for image generation (see Table 1). It improves over the base Janus-Pro-7B by 0.17, with the largest increases obtained from *Counting* (+0.34), *Position* (+0.21), and *Attribute Binding* (+0.27). Notably, MILR surpasses frontier non-reasoning models such as SD3-Medium, BAGEL, and GPT-4o (+12%). Compared with training-based reasoning models (e.g., GoT-R1 and T2I-R1), MILR performs better and requires no parameter tuning. For fairness, we also compare MILR with test-time reasoning models. Surprisingly, it outperforms ReflectionFlow and PARM (+4.5%) that rely on scaling up test-time computation, demonstrating the superiority of our test-time optimization method.

We further evaluate MILR on two additional benchmarks: T2I-CompBench and WISE (see Table 2). Again, it achieves the best performance on both, highlighting the robustness of our model. Specifically, on T2I-CompBench, MILR improves over the base Janus-Pro-7B by a large margin (+0.14)

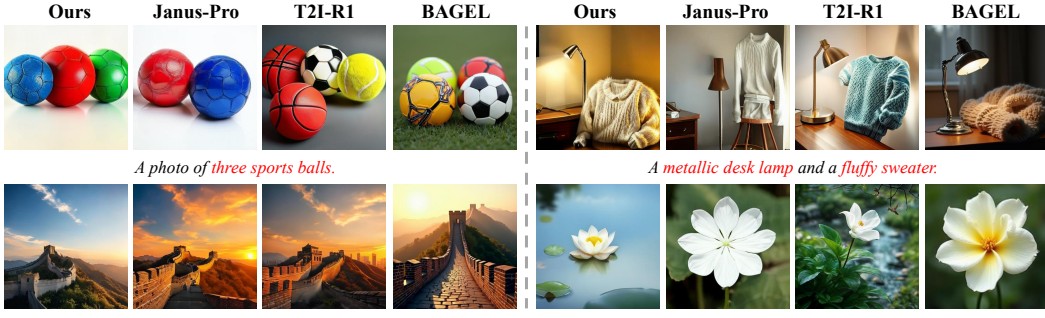

| Ours | Janus-Pro | T2I-R1 | BAGEL | Ours | Janus-Pro | T2I-R1 | BAGEL |

*A photo of three sports balls.*          *A metallic desk lamp and a fluffy sweater.*

*The Great Wall of China when it's 3 PM in Los Angeles. [6 AM in China]*     *A flower that symbolizes purity in China. [Referring to lotus]*

Figure 3: Qualitative studies on theree benchmarks. Reasoning cues are highlighted in red.

Table 3: Ablations of MILR on GenEval, T2I-CompBench, and WISE

| Method | GenEval | | | | | | | T2I-CompBench | WISE |
|---|---|---|---|---|---|---|---|---|---|
| | Single Obj. | Two Obj. | Counting | Color | Pos. | Attr. Binding | Overall | Overall | Avg |
| Janus-Pro-7B+MILR (ours) | 1.00 | 0.96 | 0.90 | 0.98 | 0.98 | 0.91 | **0.95** | **0.5325** | **0.63** |
| w/o MILR | 0.98 | 0.85 | 0.56 | 0.89 | 0.77 | 0.64 | 0.78 | 0.3921 | 0.35 |
| w/o Image | 1.00 | 1.00 | 0.91 | 0.95 | 0.95 | 0.88 | 0.94 | 0.5210 | 0.61 |
| w/o Text | 1.00 | 0.95 | 0.88 | 0.91 | 0.97 | 0.89 | 0.93 | 0.5043 | 0.56 |

and slightly outperforms T2I-R1, a strong training-based reasoning model. On WISE, which emphasizes world knowledge understanding, MILR outperforms the base Janus-Pro-7B (+80%) and the second-best model T2I-R1 (+16.7%), implying the importance of reasoning in comprehending knowledge-intensive instructions (see our qualitative studies of reasoning trajectories in Figure 8).

**MILR is capable of geometric, temporal, and cultural reasoning.** Next, we conduct a qualitative study on knowledge-intensive WISE. Surprisingly, MILR demonstrates nontrivial ability in geometric, temporal and cultural reasoning (see Figure 3). Take the prompt "The Great Wall of China when it's 3 PM in Los Angeles", MILR correctly infers the time difference from its geometric knowledge of China and Los Angeles, and concludes that "The Great Wall at Dawn" is of interest. As another example, MILR correctly infers that lotus symbolizes purity in Chinese culture.

**Joint image-text reasoning in the unified latent space leads to the best performance.** To better understand the contribution of each modality to the strong performance of MILR, we individually ablate the latent optimization of images and text, denoted by "w/o image" and "w/o text", respectively (see Table 3). First, we find that both of them exceed the base model (w/o MILR) by a large margin (e.g., > 0.21 on WISE), and optimizing both modalities leads to the best performance. Interestingly, text-only optimization (w/o image) fares slightly better than image-only optimization (w/o text), and approaches the performance of our best model (+MILR) on all benchmarks, suggesting substantial room for improvement in the language understanding component of MUG-based image generation models.

## 4.3 ANALYSIS

### 4.3.1 HYPERPARAMETERS

We analyze three important hyperparameters of MILR: (1) the maximum optimization step $T$, (2) the portion of text tokens to be optimized $\lambda_t$ in text-only optimization, and (3) the proportion of image tokens to be optimized $\lambda_v$ in image-only optimization. We perform analysis 1 on all three benchmarks and analyses 2 and 3 on a held GenEval validation split. For each setting, we run MILR three times with different random seeds and report the average scores.

**Scaling up the number of optimization steps improves performance on all benchmarks.** MILR admits test-time compute scaling by increasing the number of optimization steps. We illustrate and compare three setups: MILR, text-only optimization, and image-only optimization (see Figure 4). On all benchmarks, increasing the number of optimization steps leads to consistent improvements, with the best performance achieved at step 16, after which model performance plateaus. Moreover, among the three setups, optimizing both images and text (i.e., MILR) yields the best per-

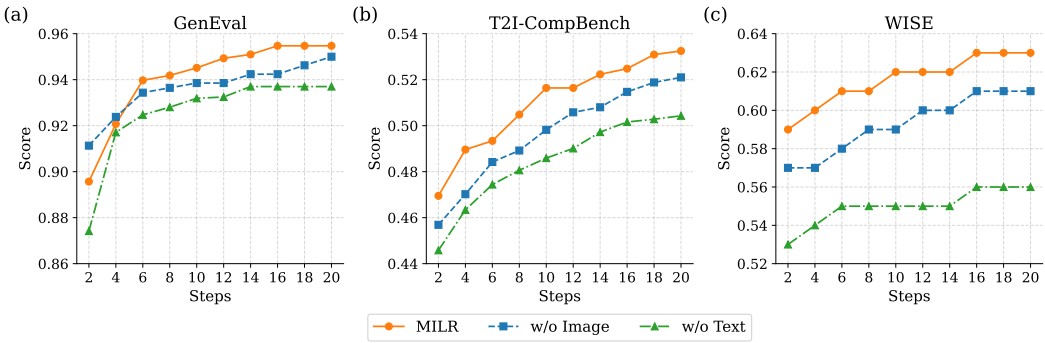

Figure 4: Performance across three benchmarks for varying optimization steps.

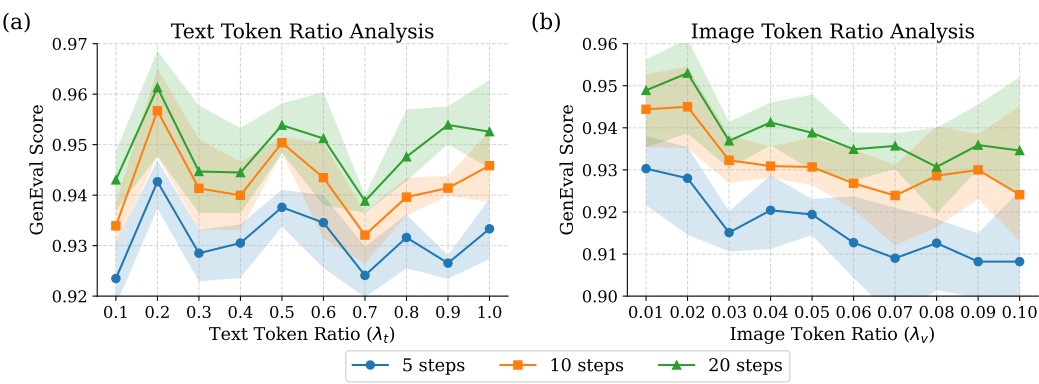

Figure 5: GenEval scores with varying optimization ratios of text and image tokens.

formance for almost all steps, except for a few early steps on GenEval, suggesting that MILR is well-suited for test-time compute scaling.

**Optimizing a moderate amount (e.g., 20%) of text tokens leads to the best performance.** When varying $\lambda_t$ from 0.1 to 1.0, text-only optimization fluctuates between 0.91 and 0.96, and reaches the highest score at $\lambda_t = 0.2$ (see Figure 5 (a)). We observe the same trend for different numbers of optimization steps. All of these suggest that MILR is relatively robust to text optimization. Moreover, we empirically find that the optimal $\lambda_t = 0.2$ corresponds to the prompts that elicit more coherent reasoning (see examples in Figure 10).

**Optimizing a tiny amount (e.g., 2%) of image tokens gives rise to the best result.** Hu et al. (2025) have empirically shown that, in autoregressive image generation, early-stage tokens govern the overall image structure, and perturbing the first 20% of image tokens results in significant structural deviations. Therefore, we conservatively adopt a very small $\lambda_v$, varying it from 0.01 to 0.1 (see Figure 5 (b)). Surprisingly, optimizing only the first 2% of all image tokens already yields peak performance, while further increasing $\lambda_v$ tends to degrade it (see examples in Figure 12).

## 4.4 REWARD MODELS

The reward model is a crucial component of MILR, providing learning signals for latent reasoning. Following previous work (Liu et al., 2025a; Tong et al., 2025), we have used the benchmark's evaluator as the reward model (denoted by 👑 *OracleReward*), but in real-world scenarios, oracle rewards are usually unknown, and it is difficult to design domain-specific rewards. To show that MILR is effective without the reliance on 👑, we test it with a set of off-the-shelf reward models on GenEval:

- 🐚 *SelfReward* uses MUG itself (e.g., Janus-Pro in this work) to evaluate images.
- 🖇 *GPT-4o* represents a frontier critic for assessing image quality (Hurst et al., 2024).
- 🔗 *UnifiedReward* is specifically tuned for a unified evaluation of MUG (Wang et al., 2025).

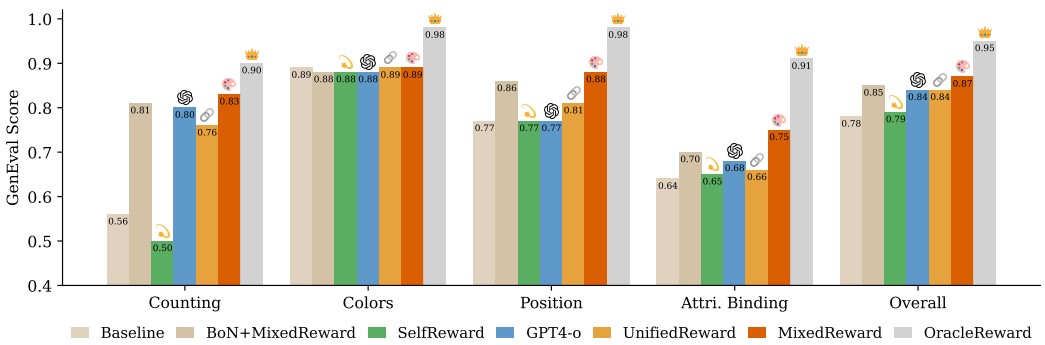

Figure 6: Performance with different reward models on GenEval.

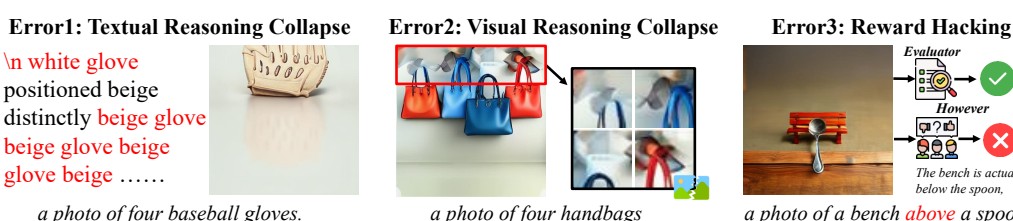

Figure 7: Error Case Study

- 🎨 *MixedReward* is a composite critic for more comprehensive evaluation. It aggregates rewards from specialized models, including GroundingDINO (Liu et al., 2024) (evaluating object detection), GIT (Wang et al., 2022) (judging colors), and 🔗 (assessing aesthetics).

Unsurprisingly, 👑 gives rise to the best performance across all dimensions (see Figure 6). For non-oracle critics, all variants surpass the baseline in terms of the overall score. Notably, MILR remains relatively robust to different reward models, except for 🍂, which performs poorly on *Counting* (around 0.5). Among non-oracle critics, 🎨 performs the best, suggesting that, in the absence of oracle rewards, we can derive a strong universal reward model by combining specialized critic models. Moreover, MILR+🎨 slightly outperforms the strong Best-of-N+🎨 baseline (+2.4%) under comparable computation (i.e., N = T = 20), once again demonstrating the superiority of our method.

### 4.4.1 ERROR CASES

Though MILR has shown the best performance across all three benchmarks, we observe three major failure modes and illustrate them respectively in Figure 7. Specifically, the failure modes include (1) **Textual Reasoning Collapse.** The regenerated chain of thoughts degenerates into repetitive phrases and becomes nonsensical for guiding image generation (e.g., repeated "beige glove"). (2) **Visual Reasoning Collapse.** While textual reasoning is fine, visual reasoning degrades prefix image tokens that govern the overall structure and subsequently adversely affects the generation of remaining tokens that control fine-grained details (e.g., blurred handles). (3) **Reward Hacking.** Models exploit shortcuts to achieve high rewards, but the generated image does not align with the given instruction perfectly (e.g., unmatched position relationship). This implies that the benchmark's evaluator is limited in that it is not yet fully capable of spatial reasoning.

## 5 DISCUSSION

We acknowledge two primary limitations of MILR. First, we have focused on an implementation of MILR that builds upon autoregressive MUG, where both text and image tokens are generated autoregressively. Another strong paradigm of MUG is through diffusion image generation, where image tokens within the same image can attend to each other (Deng et al., 2025). It is interesting to see if the effectiveness of MILR transfers. Second, MILR relies on a reward model for learning signals, but, in practice, a perfect reward model usually does not exist, and it is difficult to design a domain-agnostic reward model. In addition, our experimental results have revealed that the strongest non-oracle reward model still lags behind the oracle reward. Thus, future work is well-suited for designing reward models that can generalize like the unified reward model of Wang et al. (2025).

## 6 CONCLUSION

We have proposed MILR, a test-time latent reasoning method for multimodal image generation. MILR employs policy gradient optimization, guided by an image quality metric, to search over the latent vector representations of discrete image and text tokens, leading to a unified multimodal reasoning framework. We implement MILR with a unified multimodal understanding and generation model that natively supports language reasoning before image synthesis. Across three benchmarks, MILR achieves state-of-the-art results. We further perform an in-depth analysis; we find that jointly reasoning in the latent image-text space is the key to its strong performance. Our qualitative studies also highlight MILR's nontrivial capability in temporal and cultural reasoning.

## 7 ETHICS STATEMENT

We strictly adhere to the ICLR Code of Ethics. Below we elaborate on the ethical considerations relevant to our work and the measures we have put in place.

**Safety of content.** To reduce potential risks, we confine our text prompts to those of the standard benchmarks, avoiding sensitive and offensive input. We manually reviewed the generated images and text used in the work and did not find harmful content.

**Privacy and intellectual property.** Our work does not involve processing personally identifiable information, as we experiment exclusively on publicly available benchmarks and off-the-shelf models, as per their respective licenses.

## 8 REPRODUCIBILITY STATEMENT

To ensure reproducibility, we present experimental details, including settings and hyperparameter configurations, in Section 4.1 and release our code at https://github.com/spatigen/milr.

## 9 ACKNOWLEDGEMENTS

Yanpeng Zhao acknowledges the support of the National Natural Science Foundation of China (12574467).

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

# A APPENDIX

## A.1 OPTIMIZATION ALGORITHMS

By default, MILR jointly optimizes text latent $\mathbf{z}^{(t)}$ and image latent $\mathbf{z}^{(v)}$ in each gradient descent step, which is denoted as MILR-Joint. As an alternative, we can optimize $\mathbf{z}^{(t)}$ and $\mathbf{z}^{(v)}$ in an alternating fashion (denoted by MILR-Alt), akin to the strategy used in coordinate descent. We further propose another variant of MILR that first optimizes $\mathbf{z}^{(t)}$ to its approximate optimum, followed by optimizing $\mathbf{z}^{(v)}$ to its approximate optimum (denoted by MILR-T2V). We illustrate the three algorithms in Algorithm 1. Their performance on GenEval is presented in Table 4. We do not see significant differences among them, so we use MILR-Joint by default because of its simplicity.

---

**Algorithm 1** MILR

---

**Require:** Instruction $c$, Learning rate $\eta$, MUG model $p$, reward threshold $\tau$, text and image fraction $\lambda_t, \lambda_v \in (0, 1]$, optimization steps $K$, Reasoning strategy $\mathcal{S} \in \{\text{JOINT}, \text{ALT}, \text{T2V}\}$

$\quad \mathbf{z}, \mathbf{t}, \mathbf{v} \leftarrow p(\mathbf{t}, \mathbf{v}|c)$ ▷ Initial latent vectors

$\quad V_f \sim p(\cdot|\mathbf{v})$

$\quad r \leftarrow R(V_f, c)$ ▷ Reward Calculation

$\quad \mathbf{z}^0 \leftarrow (z_{1:\lambda_t|\mathbf{t}|}^{(t)}; z_{1:\lambda_v|\mathbf{v}|}^{(v)})$ ▷ Set $\lambda_t$ and $\lambda_v$ fraction

$\quad k \leftarrow 1$ ▷ Starting step index

$\quad$ **while** $k \leq K$ and $r \leq \tau$ **do**

$\quad\quad V_f, \mathbf{z}^k \leftarrow LatentReasoning_{\mathcal{S}}(\mathbf{z}^{k-1}, \eta, k, K, c, p)$ ▷ Select a strategy: Algorithm 2

$\quad\quad r \leftarrow R(V_f, c)$

$\quad\quad k \leftarrow k + 1$

$\quad$ **end while**

$\quad$ **return** $V_f$

---

---

**Algorithm 2** LatentReasoning (default: JOINT)

---

**(a)** JOINT

1: $\mathbf{z}^k \leftarrow \mathbf{z}^{k-1} + \eta \cdot \mathcal{J}(\mathbf{z}^{k-1})$
2: $V_f \sim p(V_f \mid \mathbf{z}^k, c)$
3: **return** $V_f, \mathbf{z}^k$

**(b)** ALT

1: **if** $k \bmod 2 = 1$ **then**
2: $\quad \mathbf{z}_k^{(t)} \leftarrow \mathbf{z}_{k-1}^{(t)} + \eta \cdot \mathcal{J}(\mathbf{z}_{k-1}^{(t)})$
3: $\quad T_f \sim p(T_f|\mathbf{z}_k^{(t)}, c)$
4: $\quad V_f \sim p(V_f|\mathbf{z}_{k-1}^{(v)}, T_f, c)$
5: **else**
6: $\quad \mathbf{z}_k^{(v)} \leftarrow \mathbf{z}_{k-1}^{(v)} + \eta \cdot \mathcal{J}(\mathbf{z}_{k-1}^{(v)})$
7: $\quad V_f \sim p(V_f|\mathbf{z}_k^{(v)}, T_f, c)$
8: **end if**
9: **return** $V_f, \mathbf{z}^k$

**(c)** T2V

1: **if** $k \leq \lfloor K/2 \rfloor$ **then**
2: $\quad \mathbf{z}_k^{(t)} \leftarrow \mathbf{z}_{k-1}^{(t)} + \eta \cdot \mathcal{J}(\mathbf{z}_{k-1}^{(t)})$
3: $\quad T_f \sim p(T_f|\mathbf{z}_k^{(t)}, c)$
4: $\quad V_f \sim p(V_f|\mathbf{z}_{k-1}^{(v)}, T_f, c)$
5: **else**
6: $\quad \mathbf{z}_k^{(v)} \leftarrow \mathbf{z}_{k-1}^{(v)} + \eta \cdot \mathcal{J}(\mathbf{z}_{k-1}^{(v)})$
7: $\quad V_f \sim p(V_f|\mathbf{z}_k^{(v)}, T_f, c)$
8: **end if**
9: **return** $V_f, \mathbf{z}^k$

---

Table 4: Geneval results with different reasoning strategies.

| Algorithms | Single Obj. | Two Obj. | Counting | Colors | Position | Attr. Binding | **Overall** |
|---|---|---|---|---|---|---|---|
| MILR-JOINT | 1.00 | 0.96 | 0.90 | 0.98 | 0.98 | 0.91 | 0.95 |
| MILR-ALT | 1.00 | 0.98 | 0.87 | 0.95 | 0.97 | 0.88 | 0.94 |
| MILR-T2V | 1.00 | 0.96 | 0.95 | 0.95 | 0.96 | 0.89 | 0.95 |

Table 5: Geneval results under random subset optimization of image tokens.

| Algorithms | Single Obj. | Two Obj. | Counting | Colors | Position | Attr. Binding | **Overall** |
|---|---|---|---|---|---|---|---|
| MILR (Image+Prefix) | 1.00 | 0.95 | 0.88 | 0.91 | 0.97 | 0.89 | 0.93 |
| MILR (Image+Random) | 1.00 | 0.92 | 0.84 | 0.89 | 0.94 | 0.86 | 0.90 |

## A.2 Qualitive Study

### A.2.1 Reasoning trajectories

MILR admits multi-round latent-space refining at test time. In Figure 8, we visualize the full trajectories of rounds 2, 3, and 4 that finally arrive at the correct generation.

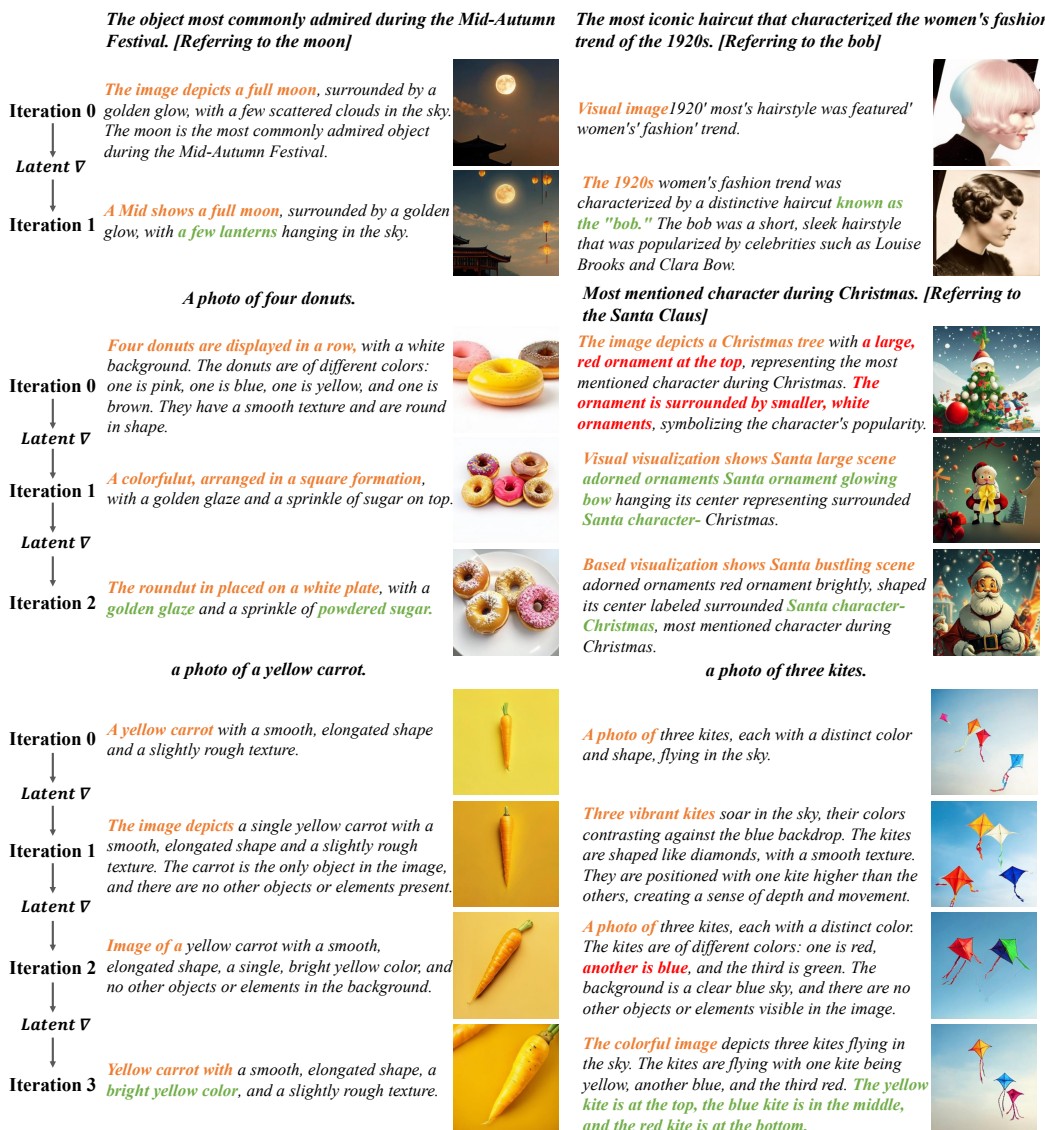

Figure 8: Example reasoning trajectories. Orange indicates prefix optimization, red highlights incorrect reasoning, and green represents correct reasoning.

### A.2.2 Model Ablations

We provide a qualitative study accompanying the model ablations done in Section 4.2. Consistent with the quantitative results, single-modality optimization surpasses the base model but underperforms joint optimization. Clearly, text-only optimization excels at tasks that rely on nontrivial numerical and compositional reasoning (e.g., "a photo of four clocks/bowls/knives" or "three kites"), while image-only optimization tends to refine image structures by adjusting the spatial arrangement of objects (e.g., "a photo of a tie above a sink"). To illustrate this scenario, consider the prompt "a photo of four clocks" in Figure 9, text-only optimization produces four spatially uncorrelated clocks,

whereas image-only optimization tends to piece together four visually similar clocks on the same wall. Interestingly, across all examples, image-only optimization exhibits a tendency to zoom in on the objects of interest.

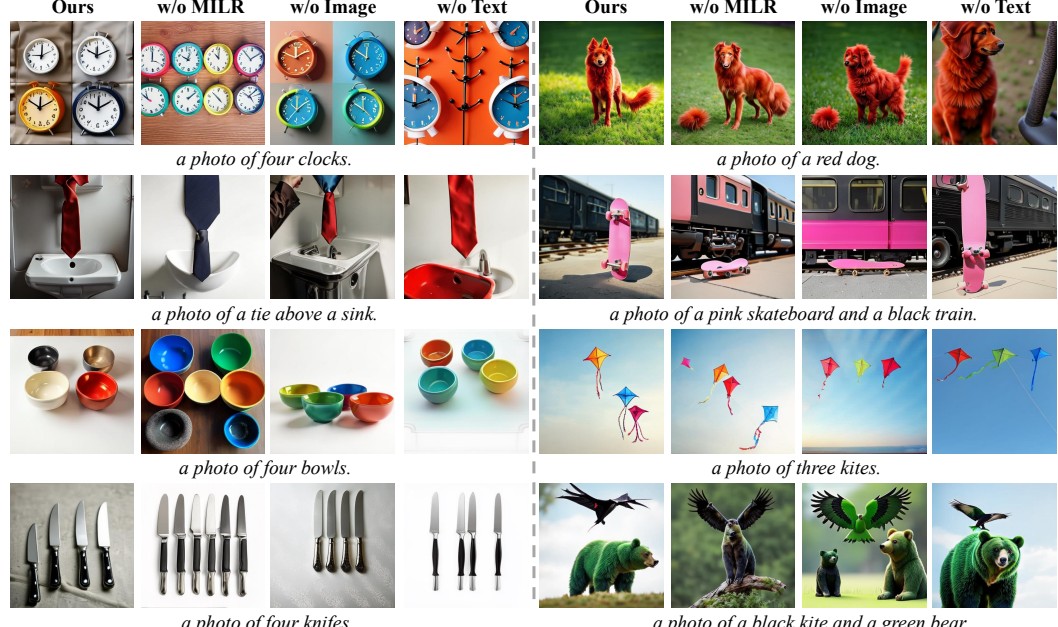

Figure 9: Example generations of MILR (ours), the base model (w/o MILR), text-only (w/o Image) optimization, and image-only (w/o Text) optimization.

## A.3 ADDITIONAL ANALYSIS

### A.3.1 ANALYSIS OF TEXT AND IMAGE OPTIMIZATION RATIOS

In Section 4.3.1, we studied how the text-token ratio $\lambda_t$ and the image-token ratio $\lambda_v$ affect MILR. We also provide example language and image reasoning trajectories when varying $\lambda_t$ in Figure 10 and Figure 11, respectively. In Figure 12, we show how image generation evolves when varying $\lambda_v$. We empirically find that $\lambda_t = 0.2$ and $\lambda_v = 0.02$ generally lead to better generation.

### A.3.2 ANALYSIS OF REWARD MODELS

In Section 4.4, we evaluated MILR with different reward models: 🌙 SelfReward, 🌀 GPT-4o, 🔗 UnifiedReward, 🎨 MixedReward, and 👑 OracleReward (see Table 6, Table 7 and Table 8). We further provide example images generated under each setup in Figure 13. Overall, 👑 establishes an upper bar of generation quality, and 🎨, among non-oracle reward models, performs best because it incorporates multiple specialized critics to provide a comprehensive assessment. Though 🎨 still lags behind 👑, in the absence of 👑, we can compose a strong universal reward model from specialized off-the-shelf image critics.

Table 6: GenEval results with different rewards. The best performance is in bold.

| Reward | Single Obj. | Two Obj. | Counting | Colors | Position | Attr. Binding | **Overall** |
|---|---|---|---|---|---|---|---|
| Baseline | 0.98 | 0.85 | 0.56 | 0.89 | 0.77 | 0.64 | 0.78 |
| 🌙 SelfReward | **1.00** | 0.90 | 0.50 | 0.88 | 0.77 | 0.65 | 0.79 |
| 🌀 GPT-4o | 0.98 | **0.94** | 0.80 | 0.88 | 0.77 | 0.68 | 0.84 |
| 🔗 UnifiedReward | **1.00** | 0.92 | 0.76 | 0.89 | 0.81 | 0.66 | 0.84 |
| 🎨 MixedReward | **1.00** | 0.90 | **0.83** | **0.89** | **0.88** | **0.75** | **0.87** |
| 👑 OracleReward | 1.00 | 0.96 | 0.90 | 0.98 | 0.98 | 0.91 | 0.95 |

**Prompt:** a photo of an apple and a donut.

**Initial Text Reasoning:** The apple is red and round, with a smooth texture. The donut is brown, round, and has a slightly rough texture. They are placed next to each other on a white plate.

**Latent Optimization:**

$\lambda_t = 0.1$    *Visual vibrant depicts a* orange apple and a chocolate donut. *<eos>*

$\lambda_t = 0.2$    *Visual vibrant depicts a orange with don donut displays* a red apple and a golden brown donut. *<eos>*

$\lambda_t = 0.3$    *Visual vibrant depicts a orange with don donut displays\n image has round with* a hole in the center, and a bright red apple with a smooth skin. *<eos>*

$\lambda_t = 0.4$    *Visual vibrant depicts a orange with don donut displays\n image has round with has with while smooth green* apple. *<eos>*

$\lambda_t = 0.5$    *Visual vibrant depicts a orange with don donut displays\n image has round with has with while smooth green, and It appleuts <eos>*

$\lambda_t = 0.6$    *Visual vibrant depicts a orange with don donut displays\n image has round with has with while smooth green, and It appleuts has golden and with with* donut. *<eos>*

$\lambda_t = 0.7$    *Visual vibrant depicts a orange with don donut displays\n image has round with has with while smooth green, and It appleuts has golden and with with with covered powdered crumb raised . <eos>*

$\lambda_t = 0.8$    *Visual vibrant depicts a orange with don donut displays\n image has round with has with while smooth green, and It appleuts has golden and with with with covered powdered crumb raised surface with The sit next* to each other. *<eos>*

$\lambda_t = 0.9$    *Visual vibrant depicts a orange with don donut displays\n image has round with has with while smooth green, and It appleuts has golden and with with with covered powdered crumb raised surface with The sit next side to one other . <eos>*

$\lambda_t = 1.0$    *Visual vibrant depicts a orange with don donut displays\n image has round with has with while smooth green, and It appleuts has golden and with with with covered powdered crumb raised surface with The sit next side to one other. what plate surface with The* apple and donut are placed on. *<eos>*

Figure 10: Case study of the text token optimization ratio $\lambda_t$. The underlined text is decoded from the optimized latents. *<eos>* denotes the end-of-sentence token.

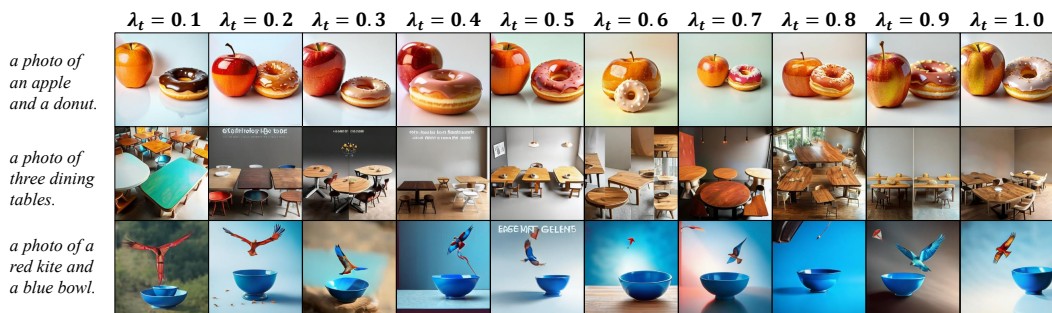

Figure 11: Generated images with varying token optimization ratio $\lambda_t$.

Table 7: T2I-CompBench results with different reward models. The best performance is in bold.

| Reward | Color | Shape | Texture | Spatial | Non-Spatial | Complex | **Overall** |
|---|---|---|---|---|---|---|---|
| Baseline | 0.6359 | 0.3528 | 0.4936 | 0.2061 | 0.3085 | 0.3559 | 0.3921 |
| 🔄 Self Reward | 0.7196 | 0.4620 | 0.5887 | 0.2379 | 0.3055 | 0.3868 | 0.4501 |
| 🌀 GPT-4o | 0.7624 | 0.4701 | 0.6561 | 0.2778 | **0.3102** | **0.3881** | 0.4775 |
| 🔗 UnifiedReward | 0.8208 | 0.4609 | 0.5835 | 0.3210 | 0.3044 | 0.3700 | 0.4651 |
| 🎨 MixedReward | 0.8009 | 0.4765 | 0.6608 | 0.4077 | 0.3055 | 0.3745 | 0.5043 |
| 👑 OracleReward | **0.8508** | **0.5117** | **0.6949** | **0.4613** | 0.3078 | 0.3684 | **0.5325** |

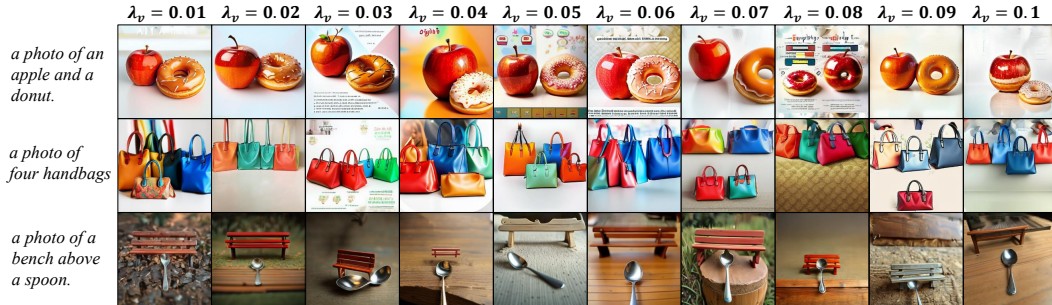

Figure 12: Generated images with varying image token optimization ratio $\lambda_v$.

Table 8: WISE results with different reward models. The best performance is in bold.

| **Reward** | Cultural | Time | Space | Biology | Physics | Chemistry | **Overall** |
|---|---|---|---|---|---|---|---|
| Base | 0.30 | 0.37 | 0.49 | 0.36 | 0.42 | 0.26 | 0.35 |
| 🍂 Self Reward | 0.40 | 0.43 | 0.53 | 0.38 | 0.44 | 0.23 | 0.41 |
| ⑤ GPT-4o | 0.53 | 0.53 | 0.62 | 0.52 | 0.53 | 0.34 | 0.52 |
| 🔗 UnifiedReward | 0.45 | 0.54 | 0.60 | 0.46 | 0.52 | 0.30 | 0.48 |
| 🎨 MixedReward | 0.48 | 0.51 | 0.61 | 0.44 | 0.51 | 0.33 | 0.49 |
| 👑 Metric Reward | **0.64** | **0.65** | **0.72** | **0.66** | **0.71** | **0.37** | **0.63** |

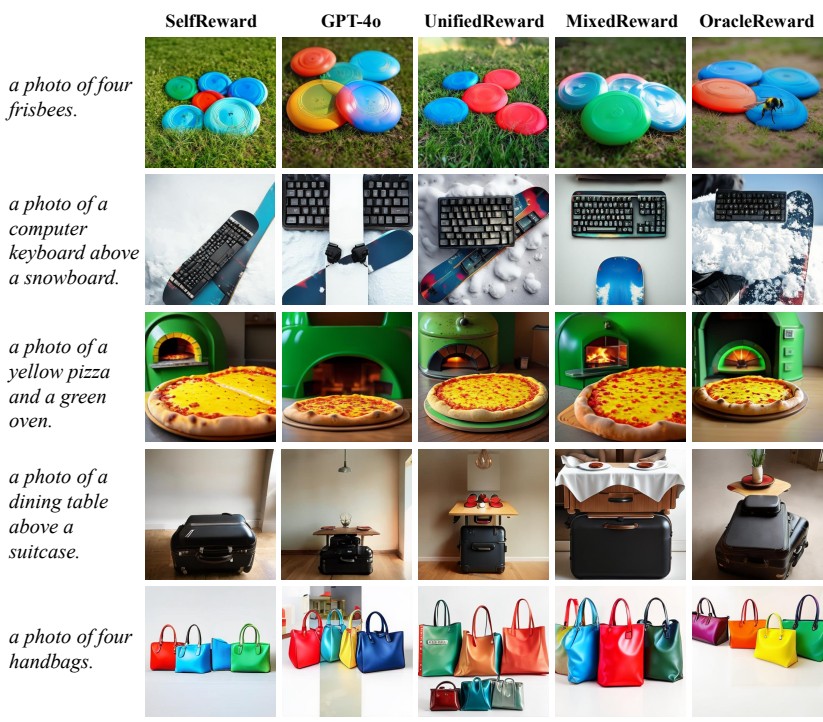

Figure 13: Case study of various reward models.

### A.3.3 EFFICIENY ANALYSIS

We summarize the computing resources used by different image generation models in Table 9. Compared with training-based reasoning approaches such as T2I-R1 (Jiang et al., 2025a), Janus-Pro with

DPO/GRPO (Tong et al., 2025), and Flow-GRPO (Liu et al., 2025a), our test-time reasoning method achieves the best performance, without relying on curated reasoning data for training. As an example, Flow-GRPO requires about 2K A800 GPU hours for training but only matches MILR, not to mention its labor cost in curating reasoning data. Moreover, when compared with the strong test-time reasoning method Best-of-N (N=20 vs. T=20 in MILR), MILR surpasses it by 0.04 while requiring less inference time because it employs an early-stop strategy, i.e., it stops once the generated image satisfies evaluation criteria.

Table 9: Efficiency comparisons.

| Method | #GPU ↓ | GPU | Training Time ↓ | Inference Time (s/sample) ↓ | Training Data | GenEval Score ↑ |
|---|---|---|---|---|---|---|
| T2I-R1 (Jiang et al., 2025a) | 8 | H800 | 16 h | – | ✓ | 0.79 |
| Janus-Pro+GRPO (Tong et al., 2025) | 8 | A100 | ∼9 h | – | ✓ | 0.81 |
| Janus-Pro+DPO (Tong et al., 2025) | 8 | A100 | ∼9 h | – | ✓ | 0.83 |
| Flow-GRPO (Liu et al., 2025a) | 24 | A800 | ∼100 h | – | ✓ | **0.95** |
| Reflect-DiT (Li et al., 2025b) | – | A6000 | 24 h | 104.1 | ✓ | 0.81 |
| Janus-Pro-7B(+Best-of-N, N=20) | 1 | A100 | 0 | 52.07 | ✗ | 0.91 |
| MILR (ours) | 1 | A100 | 0 | 32.54 | ✗ | **0.95** |

### A.4 LATENT REGULARIZATION TO MITIGATE REWARD HACKING

As discussed in Section 4.4.1, MILR can suffer from reward hacking, which refers to a scenario where a MUG model achieves high rewards but exhibits low image generation quality. To mitigate this issue, we adopt a simple $\ell_2$ regularizer to constrain the difference between the latents before and after optimization. Let $\mathbf{z}_{\text{init}}$ denote the initial latents produced by a frozen MUG model before optimization, and let $\mathcal{J}(\mathbf{z}^k)$ denote the original objective used at the latents optimization step $k$ (see Equation (6)), we define the regularized objective as:

$$\mathcal{J}_{\text{reg}}(\mathbf{z}^k) = \mathcal{J}(\mathbf{z}^k) - \beta\big\|\mathbf{z}^k - \mathbf{z}_{\text{init}}\big\|_2^2, \tag{7}$$

where $\beta > 0$ controls the strength of regularization. Intuitively, this term encourages the optimized latents $\mathbf{z}^k$ to stay close to the initial latents, thus maintaining image generation capability similar to that of the frozen MUG model.

In our experiments, we use a small coefficient $\beta = 0.05$, which empirically stabilizes the optimization and alleviates semantic collapse, without degrading generation quality. In Figure 14, we illustrate that adding latent regularization helps mitigate reward hacking.

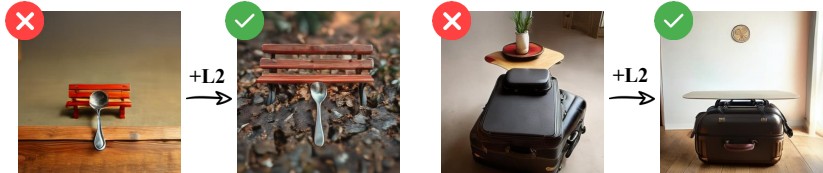

*a photo of a bench above a spoon*    *a photo of a dining table above a suitcase.*

Figure 14: $\ell_2$ regularization on latents mitigates reward hacking.

### A.5 THE USE OF LARGE LANGUAGE MODELS (LLMS)

We used LLMs (e.g., OpenAI ChatGPT) only to polish our writing and to create LaTeX algorithms and math equations. LLMs did not contribute to our research idea, model design, experimentation, etc. All contents of the paper have been verified by the authors.

