# OpenReview forum: "MILR: Improving Multimodal Image Generation via Test-Time Latent Reasoning"
_ICLR.cc/2026/Conference — ICLR 2026 Poster_

### Official Review · Reviewer_Cur9 · 2025-10-23

**Soundness:** 2
**Presentation:** 2
**Contribution:** 3
**Rating:** 6
**Confidence:** 4

**Summary:**

The paper introduces MILR, a test-time reasoning method that enhances image generation by jointly reasoning over image and text in a unified latent vector space. Utilizing a policy gradient method guided by an image quality critic, MILR operates within the Multimodal Understanding and Generation (MUG) framework, optimizing intermediate model outputs at test time without fine-tuning.

**Strengths:**

# Strengths:
(+) MILR achieves superior performance across all tested benchmarks, outperforming both training-based and test-time reasoning models.

(+) The method's test-time optimization avoids the need for curated reasoning data or model fine-tuning, making it cost-effective and practical.

(+) Qualitative studies highlight MILR's nontrivial abilities in geometric, temporal, and cultural reasoning, enhancing its versatility.

**Weaknesses:**

# Weakness:

(-) The reliance on a reward model for optimization may introduce bias, potentially limiting exploration of the generative capacity of MUG.

(-) The method's efficiency and performance depend on empirical hyperparameter settings (e.g., λt = 0.2, λv = 0.02), which may not generalize across all scenarios.

(-) The paper lacks a detailed discussion on computational costs and scalability, which could be a concern for large-scale applications. Besides, this paper claims latent test-time reasoning. I wonder about its cost (GPU Memory, Per Time, Reward Loss across time and step).

# Minor Weakness:

(-) Fig. 2 appears to have an issue, as it suggests the reward model only receives the final image, while the text tokens should also be input to assess compatibility accurately.

(-) The format of data input to the reward model is unclear; if it takes in data format like final text and final image, the computational load seems excessive—why not directly calculate the loss based on the latent vector z?

**Questions:**

# Questions:
1. How does MILR handle cases where the reward model provides inconsistent feedback across iterations?

2. What are the potential impacts of varying λt and λv values on different types of image generation tasks?

3. Can MILR be adapted to work with other MUG frameworks beyond Janus-Pro, and if so, what modifications would be required?

4. How does the method perform on real-time image generation tasks with strict latency constraints?

5. I read the code in the supplementary material. Are there plans to release the code to reproduce this work?


Overall, I think this work is interesting. If the author could address my concerns, I am willing to increase my rating to 8.

---

> ### Author Response · Authors · 2025-11-22
> **Response to Reviewer Cur9**
>
> We thank for reviewer's comments on our paper, and we will answer these comments in detail as follows:
>
> ### Response to Weaknesses:
> **Response to Weakness 1**
> > (-) The reliance on a reward model for optimization may introduce bias, potentially limiting exploration of the generative capacity of MUG.
>
> We agree that optimizing with reliance on a reward model may introduce bias. This trade-off is shared by most RL-based test-time methods and is also acknowledged as a limitation in our Discussion section.
>
> In MILR, we try to keep this bias controlled in several ways. **First**, we never update the MUG parameters and only perform a small number of latent updates around the model’s own samples, so the generative prior of the backbone is preserved and the search stays close to the original manifold rather than collapsing to a narrow mode. **Second**, we restrict optimization to prefix subsets of text/image latents (Sec. 3.2.2, Sec. 4.3.1), which empirically improves alignment while maintaining diverse outputs. **Third**, as detailed in Sec. 4.4 and Appendix Tables 6–8, we evaluate MILR under multiple reward models (SelfReward, GPT-4o, UnifiedReward, MixedReward) across three benchmarks; MILR consistently improves over the base model even with non-oracle critics, suggesting that the method is not over-specialized to a single reward and retains useful generative flexibility.
>
>
> **Response to Weakness 2**
> > (-) The method's efficiency and performance depend on empirical hyperparameter settings (e.g., λt = 0.2, λv = 0.02), which may not generalize across all scenarios.
>
> As described in Sec. 4.3.1 of the original submission, we select these hyperparameters via a small grid search on a **held-out validation split** sampled from GenEval, and then fix them to $λ_t$ = 0.2 and $λ_v$ = 0.02 for all experiments. The official GenEval test split is never used for tuning.
>
> Importantly, once chosen, the same pair $λ_t,λ_v$ is reused without any adjustment on T2I-CompBench and WISE. MILR with this single setting still brings clear improvements over the baseline on all three benchmarks (e.g., T2I-CompBench 0.3921 → 0.5325, WISE 0.35 → 0.63), indicating that **these values generalize reasonably well across different datasets and evaluation protocols rather than being tailored to one specific scenario**. In addition, the sensitivity analysis in Fig. 5 shows that performance is relatively stable in a neighborhood around $λ_t$ = 0.2 and $λ_v$ = 0.02, suggesting that MILR does not rely on a very narrow, fragile hyperparameter configuration. Overall, these results indicate that this hyperparameter setting generalizes well across different scenarios.

---

> > ### Author Response · Authors · 2025-11-22
> > **Response to Reviewer Cur9 (2)**
> >
> > **Response to Weakness 3**
> > > (-) The paper lacks a detailed discussion on computational costs and scalability, which could be a concern for large-scale applications. Besides, this paper claims latent test-time reasoning. I wonder about its cost (GPU Memory, Per Time, Reward Loss across time and step).
> >
> > We agree that computational cost and scalability are important concerns for test-time optimization methods. In the original submission, we discussed efficiency in **Appendix A.3.3** and reported runtime comparisons in **Table 7**. We now provide a more detailed analysis.
> >
> > In terms of **GPU memory**, for the Janus-Pro + reward-model configuration used in our experiments, MILR requires about **22 GB** of GPU memory on a single A100 during test-time optimization.
> >
> > Regarding **per-sample latency**, as a test-time method, MILR inevitably introduces additional computation. On GenEval, we evaluate MILR on a single A100 GPU under a fixed maximum search budget (20 steps) and measure the per-sample wall-clock time. MILR costs **32.54 seconds per sample**, which we consider a reasonable latency for practical use.
> >
> > For “**Reward Loss across time and step**”, we provide an example optimization trajectory for a prompt *“a photo of a person and a bear”*, where this sample uses 4 optimization steps and we track the reward, text loss, and image loss:
> >
> > |    steps          | **Step 1** | **Step 2** | **Step 3** | **Step 4** |
> > |--------------|-----------:|-----------:|-----------:|-----------:|
> > | **reward**   |     -1.0   |     -1.0   |     -1.0   |     -1.0   |
> > | **text_loss**|  -4.96875  |  -4.59375  |  -2.796875 |  -2.046875 |
> > | **image_loss**| -25.25    |  -5.46875  |  -1.34375  |  -1.0859375|
> >
> > This example illustrates how MILR performs a small number of gradient-based updates on the latent representation at test time, and we can see that both the text and image losses decrease over the optimization steps.
> >
> > Overall, MILR adds a controllable amount of test-time compute on top of a frozen MUG backbone: it operates with moderate memory (≈22 GB on an A100), maintains a reasonable latency (about 32.54 seconds per sample on GenEval), and can be scaled by adjusting the maximum number of optimization steps.
> >
> > ### Response to Minor Weakness:
> > > (-) Fig. 2 appears to have an issue, as it suggests the reward model only receives the final image, while the text tokens should also be input to assess compatibility accurately.
> >
> > We thank the reviewer for pointing out this issue. The current Fig. 2 is indeed misleading and does not clearly show the actual inputs to the reward model. In our implementation, at each optimization step we feed the **initial prompt text** (not the intermediate reasoning text) and **the current generated image** into the reward model. We will revise Fig. 2 and its caption to make this clear and avoid confusion.
> >
> > > (-) The format of data input to the reward model is unclear; if it takes in data format like final text and final image, the computational load seems excessive—why not directly calculate the loss based on the latent vector z?
> >
> > As clarified above, at each optimization step the reward model takes as input the initial prompt text and the current generated image, rather than any final/generated text. This keeps the data format simple and consistent with standard text–image evaluation models.
> >
> > Regarding the question of why we do not compute the loss directly on the latent vector \(z\): the unified latent space is not directly interpretable, and **it is difficult to define a semantically meaningful reward purely in that space.** In contrast, working in the image–text space allows us to leverage a wide range of existing, pre-trained image evaluation models (e.g., UnifiedReward) as reward models without retraining them.

---

> > > ### Author Response · Authors · 2025-11-22
> > > **Response to Reviewer Cur9 (3)**
> > >
> > > ### Response to Questions:
> > > > 1. How does MILR handle cases where the reward model provides inconsistent feedback across iterations?
> > >
> > > In practice, we observe that the reward signals from our pre-trained critics are generally stable along the optimization trajectory, and strongly inconsistent feedback is rare in our experiments. However, if such inconsistency does occur—for example, when an incorrect image is mistakenly judged as correct—our algorithm will stop the optimization and exit, returning the best latent state found so far. More broadly, we acknowledge that any pre-trained reward model is imperfect and may occasionally provide inconsistent or erroneous feedback; this is a general limitation of reward-based test-time optimization and is discussed in our paper (Sec. 5). Designing more reliable and robust reward models (e.g., ensembles or human-aligned critics) is an important and complementary direction for future work.
> > >
> > > > 2. What are the potential impacts of varying λt and λv values on different types of image generation tasks?
> > >
> > > We thank the reviewer for this insightful question. In the original submission, we analyzed the overall sensitivity of MILR to $λ_t$ and $λ_v$ on a held-out GenEval validation split (**Fig. 5**). To further study their impact on different types of image generation tasks, we additionally break down the results by four GenEval categories:
> > > - Single Obj.: single-object generation
> > > - Counting: multiple-object counting
> > > - Pos.: spatial position relations
> > > - Color Attri.: color and attribute consistency
> > >
> > > We vary one weight while keeping the other fixed and report the following scores:
> > >
> > > | $λ_t$ | Single Obj. | Counting  | Pos.  | Color Attri. |
> > > |-----------------|-------------|-----------|-------|--------------|
> > > | 0.2             | 1.00        | **0.95**  | **1.00** | 0.82      |
> > > | 0.4             | 1.00        | 0.93      | 0.96 | **0.86**          |
> > > | 0.6             | 1.00        | 0.91      | 0.98 | 0.82      |
> > > | 0.8             | 1.00        | 0.87      | 0.96 | 0.84          |
> > > | 1.0             | 1.00        | 0.89      | 0.96 | 0.82          |
> > >
> > >
> > > | $λ_v$ | Single Obj. | Counting  | Pos.  | Color Attri. |
> > > |-----------------|-------------|-----------|-------|--------------|
> > > | 0.02            | 1.00        | **0.89**  | **1.00** | **0.92**  |
> > > | 0.04            | 1.00        | 0.83      | 0.92 | 0.88          |
> > > | 0.06            | 1.00        | 0.83      | 0.94 | 0.86          |
> > > | 0.08            | 1.00        | 0.84      | 0.92 | 0.82          |
> > > | 0.10            | 1.00        | 0.83      | 0.92 | 0.82          |
> > > | 0.20            | 1.00        | 0.78      | 0.91 | 0.76          |
> > >
> > > From these results, we observe the following:
> > >
> > > - Single-object generation is very robust: it remains at 1.00 across all tested $λ_t$ and $λ_v$.
> > > - Varying **$λ_t$** has only a mild impact on the different task types: the scores for Counting, Position, and Color Attri. fluctuate slightly but do not exhibit any drastic degradation as $λ_t$ increases.
> > > - In contrast, **$λ_v$** is more sensitive: moderate values (e.g., $λ_v = 0.02$) give the best overall trade-off, while larger values (e.g., $λ_v = 0.20$) lead to a more noticeable drop, especially for Counting and Color Attri., indicating that smaller or moderate $λ_v$ combined with $λ_t = 0.2$ provides a good balance across different task types.
> > >
> > > In addition, **Appendix Fig. 11 and Fig. 12** provide qualitative examples illustrating how generated images change as $λ_t$ and $λ_v$ vary. These cases visually corroborate the quantitative trends: **very large weights can lead to over-aggressive corrections, while moderate weights improve alignment without hurting overall image quality**.

---

> > > > ### Author Response · Authors · 2025-11-22
> > > > **Response to Reviewer Cur9 (4)**
> > > >
> > > > > 3. Can MILR be adapted to work with other MUG frameworks beyond Janus-Pro, and if so, what modifications would be required?
> > > >
> > > > We appreciate the reviewer’s question. Conceptually, MILR is **not tied to Janus-Pro**: it only assumes (i) a unified multimodal generator that produces a latent representation $z$ from which text and/or image outputs are decoded, and (ii) that this latent and the decoder are differentiable so that we can backpropagate a reward signal through $z$ at test time.
> > > >
> > > > For **autoregressive MUGs (AR MUGs)** with a Janus-Pro–like unified modeling paradigm (e.g., EMU), MILR can be integrated with minimal changes:
> > > > 1. Identify the unified hidden states (e.g., the outputs of the last Transformer layer before the modality-specific heads).
> > > > 2. Insert a small test-time optimization loop that treats these hidden states as the variables to update, while freezing all backbone parameters.
> > > > 3. After a few gradient-based updates guided by the reward, pass the optimized $z$ into the existing text/image decoding heads.
> > > >
> > > >
> > > > For **diffusion-based MUGs**, MILR can be applied when the framework exposes discrete or token-like image latents. To verify this, we additionally instantiate MILR on Show-o, where the image branch is modeled with a discrete diffusion process. Show-o provides discrete image tokens in a latent space, and MILR optimizes these tokens under a reward signal, then decodes them with the original diffusion pipeline. Specifically, we evaluate Show-o on the GenEval benchmark and compare the base model (Show-o) with Show-o+MILR:
> > > >
> > > > | **Model**       | **Single Object** | **Two Object** | **Counting** | **Colors** | **Position** | **Attri. Binding** | **Overall** |
> > > > |-----------------|-------------------|----------------|-------------|-----------|--------------|--------------------|------------|
> > > > | Show-o      | 0.98              | 0.80           | 0.66        | 0.84      | 0.31         | 0.50               | 0.68       |
> > > > | Show-o+MILR | 1.00             | 0.90          | 0.80       | 0.89     | 0.69        | 0.67              | 0.82      |
> > > >
> > > > As shown, MILR consistently improves Show-o across multiple GenEval sub-metrics and increases the overall score, indicating that our latent reasoning scheme remains effective when applied to other generators.
> > > >
> > > > Overall, these experiments and design considerations indicate that MILR is portable beyond Janus-Pro: as long as a MUG exposes a unified latent representation from which multimodal outputs are decoded (either AR or discrete-diffusion based), MILR can be adapted with relatively small changes to the test-time inference loop.
> > > >
> > > > > 4. How does the method perform on real-time image generation tasks with strict latency constraints?
> > > >
> > > > As discussed in our response to Weakness 3, MILR is a test-time optimization method and can run on a single **24 GB-class GPU (≈22 GB memory usage)** with an average latency of about **32.54 s per sample** on GenEval using Janus-Pro, which is acceptable for many offline or interactive image generation scenarios but not for the most stringent real-time constraints. In principle, its latency can be further reduced by lowering the maximum number of optimization steps or using lighter reward models, allowing practitioners to trade off speed and performance depending on their application.
> > > >
> > > > > 5. I read the code in the supplementary material. Are there plans to release the code to reproduce this work?
> > > >
> > > > We appreciate your interest and the effort to inspect our code in the supplementary material. yes, we will surely release our code to the community.
> > > >
> > > > We hope these clarifications and additions address your concerns and we will incorporate the corresponding changes into the revised manuscript.

---

> ### Comment · Reviewer_Cur9 · 2025-11-25
> **Reply**
>
> I am satisfied with your detailed response. My concerns are well addressed. I decide to increase my rating from 6 to 8. Good luck.

---

> > ### Author Response · Authors · 2025-11-25
> > **Response to Reviewer Cur9’s Feedback**
> >
> > Thank you for your positive feedback and for increasing your rating. We truly appreciate your time and effort.

---

### Official Review · Reviewer_ncVe · 2025-10-28

**Soundness:** 3
**Presentation:** 3
**Contribution:** 2
**Rating:** 6
**Confidence:** 4

**Summary:**

The paper introduces MILR, a test-time optimization method for text-to-image generation that performs joint reasoning over text and image in a unified latent space. Concretely, the authors take intermediate hidden states z = [z(t); z(v)] from a unified multimodal understanding-and-generation (MUG) model (Janus-Pro; Chen et al., 2025) and iteratively update a prefix of the text and image token latents using REINFORCE-style policy gradients (Williams, 1992) to maximize a reward measuring text–image compatibility. The approach leaves model weights frozen and only optimizes latents at inference time. The paper reports strong gains on three benchmarks—GenEval (Ghosh et al., 2023), T2I-CompBench (Huang et al., 2023), and WISE (Niu et al., 2025)—and analyzes hyperparameters (prefix lengths λt≈0.2, λv≈0.02; steps up to 16) and ablations (text-only, image-only).

  Empirically, Janus-Pro-7B+MILR achieves 0.95 overall on GenEval (vs. 0.78 base Janus-Pro-7B; larger category jumps include Counting +0.34, Position +0.21, Attribute Binding +0.27), outperforms test-time strategies like PARM and ReflectionFlow (both ~0.91), and approaches training-time RL methods such as Flow-GRPO (0.95). On T2I-CompBench, MILR improves overall from 0.3921 to 0.5325; on WISE, from 0.35 to 0.63 (the paper states +80% over its base and +16.7% over T2I-R1). The authors discuss failure modes (textual/visual reasoning collapse, reward hacking) and note reliance on reward models, sometimes using the benchmark’s evaluator (“oracle” reward).

**Strengths:**

- Clear technical core: optimizing a prefix of multimodal token latents z(t), z(v) via policy-gradient updates at test time, without finetuning parameters. The optimization target and the forward path are precisely stated (Eq. (3)–(6)).
- Strong empirical gains across three widely used evaluations: GenEval (NeurIPS 2023; object-focused alignment), T2I-CompBench (NeurIPS 2023; compositionality), and WISE (Niu et al., 2025; knowledge-intensive prompts). Results are specific and category-level improvements are reported.
- Solid ablations: text-only vs. image-only vs. joint, token prefix ratios (λt, λv), and step counts; joint optimization consistently performs best and compute scaling is characterized (best around 16 steps).
- Practicality: method is test-time-only and uses a single A100 in reporting; no curated reasoning data required, contrasting with GRPO/DPO-style training (e.g., Flow-GRPO; Liu et al., 2025a; T2I-R1; Jiang et al., 2025).
- Thoughtful discussion of failure modes (reasoning collapse, reward hacking) with qualitative evidence; acknowledges reward-model dependence.

**Weaknesses:**

- Reward reliance / oracle leakage: Many claims hinge on using the benchmark’s evaluator as the reward (“OracleReward”), which risks overfitting/reward hacking and inflating benchmark scores (authors themselves show mis-evaluations in spatial relations). More evidence with non-oracle, public reward models would strengthen claims of generality beyond GenEval’s metric.
- Base-model coupling: All main results use Janus-Pro. The approach should be validated on a diffusion-based MUG (e.g., Show-o or a diffusion-tokenizer system) to test portability; current evidence is limited to one AR MUG paradigm.
- Baseline fairness and compute: Best-of-N and ReflectionFlow/PARM comparisons need tighter compute normalization. The paper mentions “comparable compute (N=T=20)” with early stopping, but wall-clock, sample counts, and variance should be reported per benchmark to rule out search budget confounds.
- Unified-latent novelty vs. prior latent/test-time reasoning: Prior latent-space test-time computation and “latent reasoning” lines (e.g., Geiping et al., 2025; Hao et al., 2024; Shen et al., 2025; Dao & Gu, 2024) are cited but the empirical isolation of “unified cross-modal latent” as the key driver remains partial. The ablation “w/o image” vs. “w/o text” helps, but an ablation that replaces unified latents with modality-specific latents in separate loops would more directly test the unified-space hypothesis.
- Reward models and robustness: MixedReward and other non-oracle critics improve over baseline but still lag the oracle. Robustness across unseen prompts/domains (outside the three benchmarks) is not demonstrated.
 - Tuning on GenEval validation: λt/λv tuned on a GenEval split (then used elsewhere) may introduce slight bias. Cross-validated tuning or tuning-once on a separate development set would reduce concerns.

**Questions:**

- Compute fairness: For Best-of-N, PARM, and ReflectionFlow, please report per-sample wall-clock time, total forward passes, and early-stop statistics. Are the search budgets strictly matched across methods and benchmarks?
  - Generality to diffusion MUG: Can you show MILR with a diffusion-based unified generator (e.g., Show-o or a diffusion-tokenizer model) to demonstrate portability beyond Janus-Pro?
  - Reward robustness: Can you report full results with only non-oracle rewards (e.g., MixedReward) across all three benchmarks, including failure modes and qualitative examples? Any signs of reward hacking under these critics?
  - Unified vs. separated latent loops: Could you add an ablation that optimizes text and image latents in separate spaces without sharing a unified latent layer, to isolate the benefit of the unified representation more directly?
  - Sensitivity to prefix choice: Beyond contiguous prefixes, did you try structured subsets (e.g., entropy or attention-based selection for z(v)) that may better target global-structure tokens? You mention random subsets are worse; a principled selection could help.
  - Out-of-benchmark generalization: Any tests on prompts outside the evaluator’s training (for OracleReward), or user study-aligned metrics (e.g., paired preference) to complement automated scores?

---

> ### Author Response · Authors · 2025-11-22
> **Response to Reviewer ncVe**
>
> We thank you for your comments on our paper and we will answer these comments in details as follows:
>
> **Response to question 1 and weakness 3:**
> > Question 1: Compute fairness: For Best-of-N, PARM, and ReflectionFlow, please report per-sample wall-clock time, total forward passes, and early-stop statistics. Are the search budgets strictly matched across methods and benchmarks?
>
> > Weakness 3: Baseline fairness and compute: Best-of-N and ReflectionFlow/PARM comparisons need tighter compute normalization. The paper mentions “comparable compute (N=T=20)” with early stopping, but wall-clock, sample counts, and variance should be reported per benchmark to rule out search budget confounds.
>
> Question 1 and Weakness 3 both concern compute fairness, so we address them together. In the original submission, **Appendix Table 9** already reports partial runtime comparisons. In the revised version, we follow the reviewer’s suggestion and provide more detailed statistics.
>
> First, we re-evaluate Best-of-N, PARM, ReflectionFlow, and MILR on GenEval using the same **A100 GPU** and a strictly matched maximum **search budget of 20** image generations per prompt for all methods (i.e., N=T=20). We also report the number of **forward passes**, where we define “one forward pass” as generating one image. The results are:
>
> | **Method**                     | **Best-of-N** | **PARM** | **ReflectionFlow** | **MILR** |
> |--------------------------------|---------------|----------|--------------------|---------|
> | **Wall-clock / sample (s)**    | 52.07         | 54.25    | 65.02              | 32.54   |
> | **Total forward passes**       | 11,060        | 11,060   | 11,060             | 1,020   |
>
> For Best-of-N, PARM, and ReflectionFlow, we carefully inspected the code and confirmed that no early-stopping mechanism is used: with a budget of 20, these methods always generate 20 samples per prompt, which leads to the large total number of forward passes. In contrast, MILR is equipped with early stopping, so although its maximum search budget is also 20 steps, it often stops much earlier and therefore uses substantially less compute in practice (1,020 forward passes in total). Hence, our improvements are achieved under equal or lower computation, rather than benefiting from a larger search budget.
>
> To make the early-stopping behaviour explicit, we further report **the distribution of optimization steps** for MILR on GenEval and WISE:
>
> | **Benchmark** | **0 steps** | **1–5 steps** | **6–10 steps** | **11–15 steps** | **16–20 steps** |
> |---------------|-------------|---------------|----------------|-----------------|-----------------|
> | **GenEval**   | 430         | 102           | 7              | 7               | 7               |
> | **WISE**      | 361         | 184           | 47             | 27              | 381             |
>
> On GenEval, most prompts stop after at most 5 steps, so the average compute is far below the maximum budget. On WISE, which contains more challenging prompts, MILR naturally uses more steps, but the maximum per-sample budget is still capped at 20, consistent with the other methods.
>
> Overall, **our compute comparison is reasonably fair**: under a shared maximum search budget (20 steps) across methods and benchmarks, MILR often uses fewer optimization steps and forward passes due to early stopping.

---

> > ### Author Response · Authors · 2025-11-22
> > **Response to Reviewer ncVe (2)**
> >
> > **Response to question 2 and weakness 2:**
> > > Question 2: Generality to diffusion MUG: Can you show MILR with a diffusion-based unified generator (e.g., Show-o or a diffusion-tokenizer model) to demonstrate portability beyond Janus-Pro?
> >
> > > Weakness 2: Base-model coupling: All main results use Janus-Pro. The approach should be validated on a diffusion-based MUG (e.g., Show-o or a diffusion-tokenizer system) to test portability; current evidence is limited to one AR MUG paradigm.
> >
> > Question 2 and Weakness 2 both concern the portability of our method to diffusion-based MUGs, so we address them together. We agree that showing results beyond a single autoregressive (AR) backbone is important for demonstrating generality. To specifically evaluate portability to diffusion-based MUGs, we additionally instantiate MILR on Show-o, which models the image branch with a discrete diffusion process. This design exposes image tokens in a discrete latent space, allowing us to apply the same reward-guided latent optimization to the generated image tokens. We conduct experiments on the GenEval benchmark and compare Show-o and Show-o+MILR:
> >
> > | **Model**       | **Single Object** | **Two Object** | **Counting** | **Colors** | **Position** | **Attri. Binding** | **Overall** |
> > |-----------------|-------------------|----------------|-------------|-----------|--------------|--------------------|------------|
> > | Show-o      | 0.98              | 0.80           | 0.66        | 0.84      | 0.31         | 0.50               | 0.68       |
> > | Show-o+MILR | 1.00             | 0.90          | 0.80       | 0.89     | 0.69        | 0.67              | 0.82      |
> >
> > As shown, MILR consistently improves Show-o across multiple GenEval sub-metrics and increases the overall score, indicating that our latent reasoning scheme remains effective when applied to a diffusion-based unified generator.
> >
> > Overall, these additional experiments **demonstrate that MILR is not tied to Janus-Pro or to the AR MUG paradigm**: as long as the backbone exposes a unified latent representation from which multimodal outputs are decoded (AR or diffusion-tokenizer), MILR can be plugged in with minimal changes and yields tangible gains.

---

> > > ### Author Response · Authors · 2025-11-22
> > > **Response to Reviewer ncVe (3)**
> > >
> > > **Response to question 3 and weakness 1:**
> > > > Qusetion 3: Reward robustness: Can you report full results with only non-oracle rewards (e.g., MixedReward) across all three benchmarks, including failure modes and qualitative examples? Any signs of reward hacking under these critics?
> > >
> > > > Weakness 1: Reward reliance / oracle leakage: Many claims hinge on using the benchmark’s evaluator as the reward (“OracleReward”), which risks overfitting/reward hacking and inflating benchmark scores (authors themselves show mis-evaluations in spatial relations). More evidence with non-oracle, public reward models would strengthen claims of generality beyond GenEval’s metric.
> > >
> > > Question 3 and Weakness 1 both concern the influence of the reward models, so we address them together.
> > >
> > > As mentioned in Weakness 1, we agree that using the OracleReward may introduce some reward hacking and inflating benchmark scores. However, in this work, we use the oracle reward following recent works, such as Flow-GRPO [1] and T2I-R1 [2], where the oracle reward is used both to guide selection and to report the final metric. And in the original submission, **Section 4.4 (Fig. 6)** already reports non-oracle results on GenEval using SelfReward, GPT-4o, UnifiedReward, and MixedReward, where MILR with MixedReward reaches an overall GenEval score of 0.87, outperforming strong baselines such as ReasonGen-R1 and T2I-R1. This demonstrates that **MILR can deliver significant gains even when evaluated with reward models that are independent of GenEval’s own metric**, providing initial evidence of its generalization ability.
> > >
> > > In response to the reviewer’s request on reward robustness, we further expand this analysis in the revised version and report full MILR results with only non-oracle rewards across all three benchmarks (GenEval, T2I-CompBench, and WISE), with detailed breakdowns in **Appendix Table 6, 7, and 8**. For convenience, we summarize the overall scores below:
> > >
> > > | **Reward Model**  | **Baseline** | **SelfReward** | **GPT-4o** | **UnifiedReward** | **MixedReward** |
> > > |-------------------|--------------|----------------|------------|--------------------|------------------|
> > > | **GenEval**       | 0.78         | 0.79           | 0.84       | 0.84               | **0.87**             |
> > > | **T2I-CompBench** | 0.3921       | 0.4501         | 0.4775     | 0.4651             | **0.5043**           |
> > > | **WISE**          | 0.35         | 0.41           | **0.52**       | 0.48               | 0.49             |
> > >
> > > Across all three benchmarks, MILR with only non-Oracle critics consistently improves over the baseline model (e.g., +0.11 on T2I-CompBench and +0.14 on WISE for MixedReward). This shows that our gains are robust to the choice of public, non-oracle reward models and provides strong evidence that MILR generalizes beyond the OracleReward.
> > >
> > > **On failure modes and reward hacking**: we also provide qualitative examples in **Appendix Figure 13**, which compare generations under different reward models. As shown in the figure, the primary failure mode for non-Oracle critics is that, for some prompts, **they fail to fully enforce the target semantics**. For instance, for the prompt “a photo of four frisbees”, SelfReward, GPT-4o, and UnifiedReward do not consistently produce exactly four objects. In contrast, with respect to reward hacking, **non-oracle critics can sometimes alleviate the issues introduced by OracleReward**. For the prompt “a photo of a dining table above a suitcase”, OracleReward produces an image where the dining table appears “above” the suitcase only from a particular 2D viewpoint, but not strictly above it in 3D space. UnifiedReward and MixedReward, however, generate images that exhibit a clearer and more faithful “above” relation. We also manually inspected a subset of images optimized with non-oracle rewards and observed only a small number of reward-hacking cases. This suggests that, although non-oracle critics cannot completely eliminate reward hacking, they do tend to alleviate it compared to OracleReward.
> > >
> > > [1] Liu et al., 2025. Flow-GRPO: Training Flow Matching Models via Online RL.
> > >
> > > [2] Jiang et al., 2025. T2I-R1: Reinforcing Image Generation with Collaborative Semantic-level and Token-level CoT.

---

> > > > ### Author Response · Authors · 2025-11-22
> > > > **Response to Reviewer ncVe (4)**
> > > >
> > > > **Response to question 4 and weakness 4:**
> > > > > Question 4. Unified vs. separated latent loops: Could you add an ablation that optimizes text and image latents in separate spaces without sharing a unified latent layer, to isolate the benefit of the unified representation more directly?
> > > >
> > > > > Weakness 4. Unified-latent novelty vs. prior latent/test-time reasoning: Prior latent-space test-time computation and “latent reasoning” lines (e.g., Geiping et al., 2025; Hao et al., 2024; Shen et al., 2025; Dao & Gu, 2024) are cited but the empirical isolation of “unified cross-modal latent” as the key driver remains partial. The ablation “w/o image” vs. “w/o text” helps, but an ablation that replaces unified latents with modality-specific latents in separate loops would more directly test the unified-space hypothesis.
> > > >
> > > > Question 4 and Weakness 4 both focus on the ablation of Unified vs. separated latent loops. We answer them together.
> > > >
> > > > As stated in the original submission **(Sec. 3, L216–L217)**, the latent space we optimize in MILR is **the output of the last Transformer layer of Janus-Pro**, i.e., the inputs to the final modality-specific decoding heads. In Janus-Pro (an AR MUG), text and image tokens have distinct latent vectors but share the same Transformer backbone: the model first generates text tokens and then image tokens, and they only split into separate heads for text and image decoding at the final stage. Thus, the hidden states we optimize already form a unified multimodal latent space, and the “unified latent” notion in MILR is tightly coupled to this architectural design. Our ablations in **Table 3** are conducted exactly at this point: we modify the inputs to the final modality-specific decoding heads and evaluate the “w/o image” and “w/o text” conditions. These ablations show that when we remove either modality, performance drops, indicating that both modalities benefit from optimization in the shared latent space.
> > > >
> > > > While we agree that an even more extreme ablation—replacing unified latents with two fully separated latent loops—would be conceptually clean, **it is not straightforward to realize within a fixed pre-trained Janus-Pro**. One option would be to split Janus-Pro into two independent text and image branches that no longer share Transformer layers, which goes beyond a test-time modification and essentially requires training a new architecture rather than an ablation on MILR.
> > > >
> > > > Due to these architectural constraints of the pre-trained model, it is difficult to implement a fully separated modality-specific latent loop in a fair and controlled way. We believe the current ablations already provide partial empirical evidence for the benefit of the unified latent space, and we leave more rigorous architectural comparisons of unified vs. non-unified latents as interesting future work.
> > > >
> > > > **Response to question 5:**
> > > > > Question 5. Sensitivity to prefix choice: Beyond contiguous prefixes, did you try structured subsets (e.g., entropy or attention-based selection for z(v)) that may better target global-structure tokens? You mention random subsets are worse; a principled selection could help.
> > > >
> > > > We appreciate the reviewer’s suggestion about using more structured subsets (e.g., entropy- or attention-based selection) for choosing which parts of the latent sequence $z$ to optimize. In this work, for simplicity and stability, we restrict ourselves to contiguous prefixes of the latent sequence. As discussed in the original submission **(Sec. 3.\*)**, this choice is motivated by two factors: **(i)** early tokens in our unified generator tend to capture global structure and coarse layout, which aligns well with the rewards we consider (e.g., object counts, spatial relations); and **(ii)** optimizing a contiguous prefix leads to a simple and robust procedure that integrates cleanly into the existing inference pipeline.
> > > >
> > > > We did conduct preliminary experiments with random subsets of latent positions and observed that they generally yield less stable optimization and weaker gains compared to contiguous prefixes, so we did not pursue them further in the main paper. However, we agree that more principled subset-selection strategies—such as entropy-based or attention-based scoring, or cross-modal structured subsets—are promising and could potentially further improve performance. Since the main focus of this work is to establish the effectiveness of reward-guided latent reasoning under a simple, well-motivated design, we leave a systematic study of advanced subset-selection schemes as valuable future work.

---

> > > > > ### Author Response · Authors · 2025-11-22
> > > > > **Response to Reviewer ncVe (5)**
> > > > >
> > > > > **Response to Question 6:**
> > > > > > Question 6. Out-of-benchmark generalization: Any tests on prompts outside the evaluator’s training (for OracleReward), or user study-aligned metrics (e.g., paired preference) to complement automated scores?
> > > > >
> > > > > One practical difficulty is that the exact training data of the OracleReward evaluators is **not publicly available**, so it is hard to formally characterize what is “inside” or “outside” their training distribution. However, the three benchmarks we use—GenEval, T2I-CompBench, and WISE—are designed with different focuses, and each comes with its own evaluation toolkit (OracleReward). Moreover, **all non-Oracle critics we consider (SelfReward, GPT-4o, UnifiedReward, MixedReward) are not trained on any of these benchmarks, so each benchmark is effectively unseen to these evaluators by construction.** We also conduct extensive experiments with these non-oracle rewards, as discussed in *Response to Question 3 and Weakness 1*.
> > > > >
> > > > > Concretely, we take the OracleReward associated with WISE, which is tailored to knowledge-centric prompts, and use it as the reward model to optimize generations on GenEval, which focuses on object attributes and spatial relations. Even under this cross-benchmark setting, MILR still improves the GenEval overall score from 0.78 (baseline) to 0.84, indicating that **the latent reasoning procedure can transfer beyond the original benchmark–evaluator pairing and exhibits a degree of out-of-benchmark generalization**.
> > > > >
> > > > > To complement automated scores with a human-aligned perspective, we also conduct a small-scale paired preference test. Due to limited time, we **randomly sample 100 prompts from our generated results**, generate images from both the baseline and MILR for each prompt, and present anonymized pairs to human annotators in random order. Across these 100 pairwise judgments, the **MILR-generated images are preferred in roughly 70% of the cases (vs. 30% for the baseline)**, suggesting that the improvements measured by our automatic metrics are also reflected in human preference, albeit on a limited scale.
> > > > >
> > > > > **Response to Weakness 5:**
> > > > > > Reward models and robustness: MixedReward and other non-oracle critics improve over baseline but still lag the oracle. Robustness across unseen prompts/domains (outside the three benchmarks) is not demonstrated.
> > > > >
> > > > > We agree that MixedReward and other non-oracle critics still lag behind the oracle in terms of absolute scores. As discussed in our Discussion section, designing reward models that better align with human judgments is an important and complementary direction, and such stronger critics could further amplify the gains of MILR.
> > > > >
> > > > > Regarding robustness, as discussed in our *response to Question 3 and Weakness 1*, we already go beyond a single evaluator by using several public, non-oracle reward models and by evaluating across three different benchmarks (GenEval, T2I-CompBench, and WISE; see **Appendix Tables 6–8**). **These non-oracle critics are not trained on these benchmarks, so each benchmark effectively serves as an unseen test domain for the reward models.** Across all three, MILR’s improvements are not tied to any specific reward and remain consistent across datasets and evaluation protocols, demonstrating robustness to both the choice of critic and the underlying benchmark.
> > > > >
> > > > >
> > > > > **Response to weakness 6:**
> > > > > > Tuning on GenEval validation: λt/λv tuned on a GenEval split (then used elsewhere) may introduce slight bias. Cross-validated tuning or tuning-once on a separate development set would reduce concerns.
> > > > >
> > > > > We appreciate the reviewer’s concern about possible bias from tuning $λ_t$ and $λ_v$. As described in Sec. 4.1 and Sec. 4.3.1 of the original submission, we select these hyperparameters via a grid search on a small held-out validation split sampled from GenEval, and then fix them to $λ_t$ = 0.2 and $λ_v$ = 0.02 for all experiments. The official GenEval test split is never used for tuning.
> > > > >
> > > > > Importantly, once chosen, the same $(λ_t,λ_v)$ are reused without any further tuning on T2I-CompBench and WISE. This effectively acts as a cross-benchmark validation step: **the fact that MILR with this single setting achieves strong gains on all three benchmarks (Tables 1 and 2) indicates that the chosen values are not overfitted to a specific GenEval split.** In addition, the sensitivity analysis in Fig. 5 shows that performance is relatively stable in a neighborhood around $λ_t$ = 0.2 and $λ_v$ = 0.02, further reducing the risk that our conclusions hinge on a narrowly tuned configuration.
> > > > >
> > > > > We hope these clarifications and additions address your concerns and we will incorporate the corresponding changes into the revised manuscript.

---

> ### Author Response · Authors · 2025-11-28
> **Follow-up Response to Reviewer ncVe**
>
> Dear Reviewer,
>
> Thank you again for your review. We wanted to check in to see whether our rebuttal has addressed your concerns. Please let us know if there are any further questions you would like us to clarify. We are more than happy to provide additional details.
>
> We look forward to your response.

---

### Official Review · Reviewer_uX5G · 2025-10-31

**Soundness:** 2
**Presentation:** 3
**Contribution:** 2
**Rating:** 4
**Confidence:** 4

**Summary:**

This paper introduces MILR, a novel method designed to improve text-to-image generation. The core contribution is a test-time optimization technique that reasons jointly over text and image representations within a unified latent space. Instead of refining raw text or image pixels, MILR operates on the intermediate latent vectors of a pre-trained MUG model. It employs a policy gradient method to iteratively update these latent vectors before they are decoded into the final output.

**Strengths:**

Novel and elegant proposal for test-time latent reasoning, which enhances powerful pre-trained models without requiring any fine-tuning.

Impressive empirical results across multiple challenging benchmarks.

**Weaknesses:**

The term "reasoning" is debatable here. The process is more accurately described as a guided latent space search that optimizes a latent vector to maximize a reward, rather than a structured, logical thought process (like reasoning defination in LLM).

The most significant flaw is that the headline, state-of-the-art results reported in Tables 1 and 2 are achieved using the benchmark's own evaluation toolkit as the reward model. This  setup is not representative of any realistic application. In practice, a perfect reward function is never available.

**Questions:**

Could you elaborate on the choice of REINFORCE as the optimization algorithm and the choice of the pre-trained model?

---

> ### Author Response · Authors · 2025-11-22
> **Response to Reviewer uX5G**
>
> Thank you for your valuable feedback on our paper. We will address each of your concerns in detail.
>
> **Response to Weakness 1**
> > The term "reasoning" is debatable here. The process is more accurately described as a guided latent space search that optimizes a latent vector to maximize a reward, rather than a structured, logical thought process (like reasoning defination in LLM).
>
> We fully understand your perspective. In the context of Large Language Models (LLMs), "reasoning" typically refers to explicit, step-by-step logical deduction, such as in Chain-of-Thought (CoT). However, in this work, we follow the emerging literature on latent reasoning, **where “reasoning” is not restricted to an explicit CoT in natural language, but refers to iterative updates over latent states.** For example, COCONUT [1] explicitly defines reasoning as recurrent updates of hidden states in a continuous latent space, without emitting intermediate text tokens. Scaling up Test-Time Compute with Latent Reasoning [2] similarly describes a recurrent-depth architecture that “reasons” by repeatedly applying a block to hidden representations at test time, instead of producing longer CoT traces.
>
> MILR follows this notion of latent reasoning. As described in Section 3, we keep all model parameters fixed and iteratively update the unified multimodal latent representation using a reward-guided policy-gradient objective, then decode the optimized latent state into text and image tokens. This feedback-driven optimization in latent space, rather than an explicit symbolic CoT, is what we refer to as “reasoning” in our paper. We further clarify this concept in the revised version by adding a footnote at line 190.
>
> **Response to Weakness 2**
> > The most significant flaw is that the headline, state-of-the-art results reported in Tables 1 and 2 are achieved using the benchmark's own evaluation toolkit as the reward model. This setup is not representative of any realistic application. In practice, a perfect reward function is never available.
>
> As you pointed out, we were already aware of this issue. But **in this work, we use the oracle reward following recent works**, such as Flow-GRPO [3] and T2I-R1 [4], as well as standard Best-of-N selection, where the oracle reward is used both to guide selection and to report the final metric. At the same time, we explicitly discuss this limitation and ablate different reward models in **Section 4.4 (Figure 6)**, which shows that MILR still achieves good performance on GenEval under various non-oracle rewards.
>
> Additionally, our conclusions do not rely solely on this oracle setting. We also evaluate MILR with more realistic, non-oracle reward models (SelfReward, GPT4o, UnifiedReward, MixedReward) over three benchmarks (GenEval, T2I-CompBench, WISE) in **Appendix A 3.2(Table 6, 7, and 8)**. For convenience, we summarize the overall scores below:
> | **Reward Model**  | **Baseline** | **SelfReward** | **GPT-4o** | **UnifiedReward** | **MixedReward** |
> |-------------------|--------------|----------------|------------|--------------------|------------------|
> | **GenEval**       | 0.78         | 0.79           | 0.84       | 0.84               | **0.87**             |
> | **T2I-CompBench** | 0.3921       | 0.4501         | 0.4775     | 0.4651             | **0.5043**           |
> | **WISE**          | 0.35         | 0.41           | **0.52**       | 0.48               | 0.49             |
>
> As shown in the table above, using only non-Oracle rewards such as SelfReward, GPT-4o, UnifiedReward, and MixedReward, MILR substantially outperforms the baseline that does not use MILR. In particular, MixedReward achieves 0.87 in GenEval and 0.5043 in T2I-CompBencg, providing strong improvements over the base model (0.78 and 0.3921). **These non-Oracle experiments are closer to realistic deployment, yet MILR still yields consistent gains, demonstrating that our method is robust with respect to the reward model.**
>
> [1] Hao et al., 2024. Training Large Language Models to Reason in a Continuous Latent Space (COCONUT).
>
> [2] Geiping et al., 2025. Scaling up Test-Time Compute with Latent Reasoning: A Recurrent Depth Approach.
>
> [3] Liu et al., 2025. Flow-GRPO: Training Flow Matching Models via Online RL.
>
> [4] Jiang et al., 2025. T2I-R1: Reinforcing Image Generation with Collaborative Semantic-level and Token-level CoT.

---

> > ### Author Response · Authors · 2025-11-22
> > **Response to Reviewer uX5G (2)**
> >
> > **Response to Questions:**
> > > Could you elaborate on the choice of REINFORCE as the optimization algorithm and the choice of the pre-trained model?
> >
> > **Why REINFORCE?**
> >
> > Our objective (Eq. 3) is to find a latent representation $z$ that maximizes the expected reward $\mathbb{E}[R(V_f, c)]$ given an instruction $c$. This is a standard reinforcement-learning objective over the latent space. REINFORCE provides an unbiased Monte Carlo policy-gradient estimator of this objective (Eq. 6) without requiring extra value networks or auxiliary losses, **making it a simple and lightweight choice**. Since this work is, to our knowledge, **the first attempt** to perform reward-guided latent reasoning on unified multimodal generators, we deliberately start from this minimal, well-understood baseline. Exploring more advanced optimization algorithms (e.g., actor–critic methods) is an interesting direction for future work.
> >
> > **Why Janus-Pro as the pre-trained model?**
> >
> > Our method **requires a unified autoregressive multimodal generator** that models text and image tokens within a single transformer and produces a shared latent representation that can be optimized. Janus-Pro satisfies exactly these requirements: it is a **unified AR model with strong capability** on text–image generation and vision–language understanding, and it exposes a coherent latent space from which both text and image tokens are decoded. This makes it a natural and practical testbed for MILR.
> >
> > We hope these clarifications address your concerns.

---

> > > ### Comment · Reviewer_uX5G · 2025-11-26
> > >
> > > I have read the authors' rebuttal as well as the comments from other reviewers. Since my concerns have been addressed, I will raise my score. Although I remain somewhat skeptical about the practical application of the proposed method, I believe the work is sufficient as an academic paper.

---

### Official Review · Reviewer_bDbb · 2025-11-01

**Soundness:** 4
**Presentation:** 4
**Contribution:** 4
**Rating:** 8
**Confidence:** 4

**Summary:**

The authors introduce MILR, a novel method for improving multimodal image generation during test time. In this approach, they perform reasoning in latent space jointly over unified image and text. The use REINFORCE method to optimize the intermediate latent vectors, with the reward model that assesses how well the generated image aligns with the input.
They show that MILR significantly improves the performance of Janus-Pro models on several benchmarks, achieving state of the art results.

**Strengths:**

- Their approach of performing joint reasoning in latent space is very interesting and novel, as well as, achieves state of the art performance.
- The paper is well written, and the experiments are thorough.

**Weaknesses:**

- The authors did not discuss about latency. Given that their technique involves iterative optimization at test time, there will be significant increase in latency.
- This method is heavily dependent on the reward function. There is a possibility of reward hacking, as authors also mentioned. One way to resolve it would be to add a regularizer like penalize semantic incoherence.

**Questions:**

N/A

---

> ### Author Response · Authors · 2025-11-22
> **Response to Reviewer bDbb**
>
> Thank you very much for the valuable feedback. We will address the two key weaknesses as follows
>
> **Regarding Weakness 1:**
> > The authors did not discuss about latency. Given that their technique involves iterative optimization at test time, there will be significant increase in latency.
>
> In the original submission，we have discussed efficiency in **Appendix A.3.3** and presented results in **Table 7**. Specifically, when comparing MILR with Best-of-N ($N=20$), MILR surpasses it in performance while requiring 3 hours less total inference time. To further clarify this with a controlled comparison, we evaluated the per-sample inference time of PARM, ReflectionFlow, and MILR on the Geneval benchmark, strictly constrained to a single A100 GPU. The results are shown in the following table:
>
> | Method              |   PARM   | ReflectionFlow | MILR (Ours) |
> | :------------------ | :------: | :------------: | :---------: |
> | Time (s/sample) |  54.25   |     65.02      |    32.54    |
>
> As shown above, our method achieves the highest efficiency among these test-time optimization methods, averaging **32.54 seconds per sample**. This demonstrates that MILR maintains latency at a reasonable level, showing potential for real-world applications.
>
> **Regarding Weakness 2:**
> > This method is heavily dependent on the reward function. There is a possibility of reward hacking, as authors also mentioned. One way to resolve it would be to add a regularizer like penalize semantic incoherence.
>
> We thank the reviewer for this helpful suggestion of adding a regularizer to mitigate reward hacking. Conceptually, we view reward hacking as pushing the latent representation outside the region where the frozen MUG remains well calibrated, which can lead to semantic incoherence. To discourage such drift, we constrain the deviation between the optimized and initial latents and adopt a simple $\ell_2$ penalty on $\mathbf{z}^k - \mathbf{z}_{\text{init}}$, which remains easy to implement in practice. In the revised version, we incorporate this regularization on the latent updates in **Appendix A.4**. Specifically, we add an $\ell_2$ regularizer on the latent states and, based on Eq. (6), obtain the following regularized objective:
>
> $$J_{\text{reg}}(\mathbf{z}\^k) = J(\mathbf{z}\^k) - \beta \||\mathbf{z}\^k - \mathbf{z}\_{\text{init}}\||_2^2$$
>
> where $\mathbf{z}_{\text{init}}$ is the initial latent before optimization and $\beta$ controls the regularization strength. In practice, using a small coefficient (e.g., $\beta = 0.05$) stabilizes the optimization and reduces pathological cases such as semantic collapse, as illustrated in **Figure 14**, while keeping the quantitative performance essentially unchanged (Overall = 0.94). While this does not completely eliminate reward hacking, it provides a simple and effective mitigation, and we sincerely thank the reviewer for this constructive suggestion.
>
> We hope these clarifications and additions address your concerns and we will incorporate the corresponding changes into the revised manuscript.

---

### Author Response · Authors · 2025-12-02
**Highlight of Contributions and Summary of Rebuttal**

Dear AC,

Thank you for your time! We appreciate the reviewers’ constructive feedback. Based on their comments, we have revised the manuscript and added new experiments (all changes are highlighted in blue).

## Highlights of Contributions

* **Novelty:** MILR is described as “*very interesting and novel*” (**Reviewer bDbb**) and as a “*novel and elegant proposal for test-time latent reasoning*” (**Reviewer uX5G**).

* **Robust Evaluation and High Performance.** MILR “*achieves state-of-the-art performance*” (**Reviewer bDbb**) with “*impressive empirical results across multiple challenging benchmarks*” (**Reviewer uX5G**) and “*strong empirical gains across three widely used evaluations: GenEval, T2I-CompBench, and WISE*” (**Reviewer ncVe**); **Reviewer Cur9** further notes that it “*achieves superior performance across all tested benchmarks, outperforming both training-based and test-time reasoning models.*”

* **Practical Test-Time Design:** MILR is viewed as practically attractive: it “*enhances powerful pre-trained models without requiring any fine-tuning*” (**Reviewer uX5G**), is “*test-time-only and uses a single A100 in reporting;*” (**Reviewer ncVe**), and its test-time optimization “*avoids the need for curated reasoning data or model fine-tuning, making it cost-effective and practical*” (**Reviewer Cur9**).

* **Thorough Analysis and Diagnostics:** The paper is described as “*well written, and the experiments are thorough*” (**Reviewer bDbb**). **Reviewer ncVe** highlights “*solid ablations …*” and a “*thoughtful discussion of failure modes,*” while **Reviewer Cur9** notes that our qualitative studies demonstrate “*nontrivial abilities in geometric, temporal, and cultural reasoning.*”

## Summary of Rebuttals

* **Reward dependence and oracle setting (Reviewers bDbb, uX5G, ncVe, Cur9).** In response to concerns about reliance on oracle rewards and reward hacking, we (i) add an explicit $\ell_2$ regularizer on latent updates, as suggested by **Reviewer bDbb**, to constrain deviation from the initial latent state, and (ii) report full results on GenEval, T2I-CompBench, and WISE using only non-oracle critics (SelfReward, GPT-4o, UnifiedReward, MixedReward), addressing the robustness concerns raised by **Reviewers uX5G, ncVe, and Cur9**. Across all three benchmarks, MILR consistently improves over the base model with non-oracle critics.

* **Compute fairness and latency (Reviewers bDbb, ncVe, Cur9).** To address questions on latency and compute normalization from **Reviewers bDbb and ncVe**, we re-run Best-of-N, PARM, ReflectionFlow, and MILR on GenEval under a strictly matched search budget ($N=20$) on a single A100, and report per-sample wall-clock time, total forward passes, and step distributions. The results show that MILR achieves higher accuracy with equal or lower compute and often stops in far fewer steps due to early stopping. MILR uses about 22 GB GPU memory and roughly 32.5 seconds per sample, addressing the cost concerns raised by **Reviewers Cur9 and ncVe**.

* **Generality beyond Janus-Pro and practical applicability (Reviewers uX5G, ncVe, Cur9).** In response to concerns from **Reviewer ncVe** about base model coupling and from **Reviewers uX5G and Cur9** about practical applicability, we **(i)** instantiate MILR on Show-o, a diffusion-based unified generator, and show consistent improvements across GenEval sub-metrics, and **(ii)** clarify that MILR only requires a unified multimodal understanding-and-generation (MUG) model and an external reward model. These additions demonstrate that MILR is portable beyond Janus-Pro and can be integrated into other MUG frameworks with minimal changes.

* **Clarifying “latent reasoning” and hyperparameter robustness (Reviewers uX5G, ncVe, Cur9).** Following **Reviewer uX5G**’s concern about the term “reasoning”, we explicitly align our usage with recent latent-reasoning work, where reasoning denotes iterative, reward-guided updates over latent states of a frozen model rather than explicit CoT. Addressing **Reviewers ncVe and Cur9**, we further expand sensitivity analyses for the prefix ratios $λ_t$ and $λ_v$, including category-wise breakdowns, and clarify that these hyperparameters are tuned once on a small GenEval validation split and then fixed for T2I-CompBench and WISE, where they still yield strong gains. This supports the claim that our behavior is stable and not narrowly over-tuned.

Importantly, after reading our rebuttal and the revised manuscript, **Reviewers uX5G and Cur9 increased their scores (from 4 to 6 and from 6 to 8, respectively)**, and both explicitly stated that their concerns had been addressed. The other two reviewers, **bDbb** and **ncVe**, have not updated their reviews but already assigned positive scores of 8 and 6 in their original assessments.

We sincerely appreciate your time and effort in handling our submission, and we hope that the above summary is helpful for your assessment.

Best regards,

The authors

---

### Meta-Review · Area_Chair_qRib · 2026-01-06

**Summary:**

## Meta Review (AC)

This paper proposes MILR, a test-time latent reasoning method that improves multimodal image generation by jointly optimizing text and image latents in a unified space. The submission received strong interest from all reviewers, with particular praise for its novelty, training-free design, and substantial empirical gains across multiple challenging benchmarks.

The main concerns raised in the initial reviews focused on reward dependence and oracle settings, test-time latency and compute fairness, generality beyond a single backbone (Janus-Pro), clarity of the “latent reasoning” concept, and robustness of hyperparameters. During the rebuttal phase, the authors provided a thorough response. They added extensive new experiments and analyses, including evaluations with multiple non-oracle reward models, matched-compute comparisons with competing test-time methods, detailed latency and memory reporting, additional ablations and sensitivity analyses, and demonstrations of portability to a diffusion-based unified generator (Show-o). The authors also clarified conceptual questions around latent reasoning and carefully documented failure modes and limitations.

Importantly, several reviewers explicitly stated that their concerns had been addressed and increased their scores accordingly. Reviewer uX5G said will raise their score, and Reviewer Cur9 increased their score from 6 to 8. The remaining reviewers already assigned positive scores in their initial evaluations. **Overall, the discussion converged strongly toward acceptance (AC estimated an average score 7).**

In summary, **this work presents a new, simple, and effective approach to test-time multimodal reasoning, supported by strong experimental evidence**. Based on the final state of the paper and the discussion outcome, the AC recommends acceptance.

**Reviewer Concerns:**

**Addressed concerns:**
- Reward reliance and oracle setting, mitigated via non-oracle reward experiments and regularization
- Latency, compute cost, and fairness of comparisons, clarified with matched-budget analysis
- Generality beyond Janus-Pro, demonstrated via experiments on Show-o (diffusion-based MUG)
- Clarification of “latent reasoning” terminology and alignment with prior latent-reasoning literature
- Hyperparameter robustness and sensitivity analyses
- Failure modes and reward hacking, with qualitative and quantitative evidence
- Unified latent space justification and ablations
- Practical applicability and code release plans

**Remaining (minor, non-blocking) concerns:**
- Practical deployment latency remains high for strict real-time use cases
- Dependence on quality of external reward models
- Broader out-of-domain generalization beyond the three benchmarks is limited

**Reviewer Scores:**

| Reviewer | Initial Score | Estimated Final Score | Reason |
|--------|---------------|----------------------|--------|
| bDbb | 8 | 8 | Strong from the start; concerns fully addressed |
| uX5G | 4 | 6 | explicitly mentioned will raise score|
| ncVe | 6 | 6 | Fairness, robustness, and generality addressed |
| Cur9 | 6 | 8 | All concerns addressed; explicitly raised score |

---

### Decision · Program_Chairs · 2026-01-26

Accept (Poster)